# In-host population dynamics of *Mycobacterium tuberculosis* complex during active disease

**Roger Vargas**[1,2]\*, **Luca Freschi**[2], **Maximillian Marin**[1,2], **L Elaine Epperson**[3], **Melissa Smith**[4,5], **Irina Oussenko**[5], **David Durbin**[6], **Michael Strong**[3], **Max Salfinger**[7,8], **Maha Reda Farhat**[2,9]\*

[1]Department of Systems Biology, Harvard Medical School, Boston, United States; [2]Department of Biomedical Informatics, Harvard Medical School, Boston, United States; [3]Center for Genes, Environment and Health, Center for Genes, National Jewish Health, Denver, United States; [4]Department of Genetics and Genomic Sciences, Icahn School of Medicine at Mount Sinai, New York, United States; [5]Icahn Institute of Data Sciences and Genomics Technology, New York, United States; [6]Mycobacteriology Reference Laboratory, Advanced Diagnostic Laboratories, National Jewish Health, Denver, United States; [7]College of Public Health, University of South Florida, Tampa, United States; [8]Morsani College of Medicine, University of South Florida, Tampa, United States; [9]Pulmonary and Critical Care Medicine, Massachusetts General Hospital, Boston, United States

**\*For correspondence:**
roger_vargas@g.harvard.edu (RV);
Maha_Farhat@hms.harvard.edu
(MRF)

**Competing interests:** The authors declare that no competing interests exist.

**Abstract** Tuberculosis (TB) is a leading cause of death globally. Understanding the population dynamics of TB's causative agent *Mycobacterium tuberculosis* complex (Mtbc) in-host is vital for understanding the efficacy of antibiotic treatment. We use longitudinally collected clinical Mtbc isolates that underwent Whole-Genome Sequencing from the sputa of 200 patients to investigate Mtbc diversity during the course of active TB disease after excluding 107 cases suspected of reinfection, mixed infection or contamination. Of the 178/200 patients with persistent clonal infection >2 months, 27 developed new resistance mutations between sampling with 20/27 occurring in patients with pre-existing resistance. Low abundance resistance variants at a purity of ≥19% in the first isolate predict fixation in the subsequent sample. We identify significant in-host variation in 27 genes, including antibiotic resistance genes, metabolic genes and genes known to modulate host innate immunity and confirm several to be under positive selection by assessing phylogenetic convergence across a genetically diverse sample of 20,352 isolates.

## Introduction

Tuberculosis (TB) and its causative pathogen *Mycobacterium tuberculosis* complex (Mtbc) remain a major public health threat (*World Health Organization, 2018*). Yet the majority of individuals exposed to Mtbc clear or contain the infection, and only 5–10% of those infected develop active TB disease at some point in their lifetime (*Pai et al., 2016*). While basic human immune mechanisms to Mtbc have been identified, attempts at effective vaccine development guided by these mechanisms have repeatedly failed (*Ernst, 2018*). Hence, global efforts in disease control currently focus on scale up of directly observed therapy but achieving a universal and sustained cure remains a challenge. Mtbc is an obligate human pathogen (*Gagneux, 2018*). Infection and disease involve a complex human host-pathogen interaction that is both physically and temporally heterogeneous (*Lin et al., 2014*). Consequently, all selective forces acting on Mtbc will originate within the host, and the study

of temporal dynamics of this is likely to inform antibiotic treatment (*Sun et al., 2012*) and rational vaccine design (*Ernst, 2018*).

Little is known about selection at short timescales, such as within single infections. Drug pressure may select for resistance-conferring mutations, thus an understanding of how the frequency of minor alleles changes longitudinally can inform optimal drug treatment (*Didelot et al., 2016*; *Sun et al., 2012*; *Zhang et al., 2016*). Mtbc's interaction with host immunity or metabolic pressures imposed by persistent active human infection may also exert selective pressures, the detection of which can inform vaccine design or host directed therapeutics. To elucidate these temporal dynamics, we aimed to study how genomic diversity arises in-host in Mtbc populations, employing a longitudinal sampling scheme from patients with active TB disease enriched for treatment failure and relapse.

The application of genome sequencing technologies to Mtbc isolates cultured from clinical samples has highlighted that infection consists of populations of Mtbc bacteria rather than single clones devoid of diversity (*Copin et al., 2016*; *Didelot et al., 2016*; *Lieberman et al., 2014*; *Lieberman et al., 2011*; *Marvig et al., 2015*). Differences in observed allele frequencies captured using genome sequencing (*Figure 1A*) may represent a difference in the genetic composition of the infecting population, commonly referred to as heterogeneity. Mtbc population heterogeneity might be present within a host because (1) the host is infected with multiple strains or is re-infected by a new strain (consistent with mixed infection or re-infection) or (2) genetic diversity arises within the Mtbc population during infection due to selection or drift (*Ford et al., 2012*; *Guerra-Assunção et al., 2015*; *Lieberman et al., 2016*). However, non-uniform sampling (*Trauner et al., 2017*), selection during the in vitro culture process (*Trauner et al., 2017*), laboratory contamination (*Goig et al., 2020*; *Wyllie et al., 2018*), sequencing error and mapping error all represent examples of experimental error that give rise to heterogeneity of low significance to host-pathogen interactions.

Here, we present a framework to overcome these barriers and demonstrate the use of longitudinally collected isolates, pooled sweeps of colonies cultured from sputa, to investigate true in-host diversity with implications for Mtbc treatment. We analyzed 614 paired longitudinal isolates representing 307 patients from eight studies (*Bryant et al., 2013*; *Casali et al., 2016*; *Guerra-Assunção et al., 2015*; *Trauner et al., 2017*; *Walker et al., 2013*; *Witney et al., 2017*; *Xu et al., 2018*). Many patients, despite undergoing treatment, remained culture positive at 2 months intervals or longer meeting microbiological criteria for delayed culture conversion, treatment failure or relapse (*Supplementary file 1–2*). Our sample consisted of 178 patients fulfilling these criteria, which allowed us to overcome the small sample size problem present in prior studies. We provide a proof of concept that whole-genome sequencing (WGS) can aide in predicting resistance

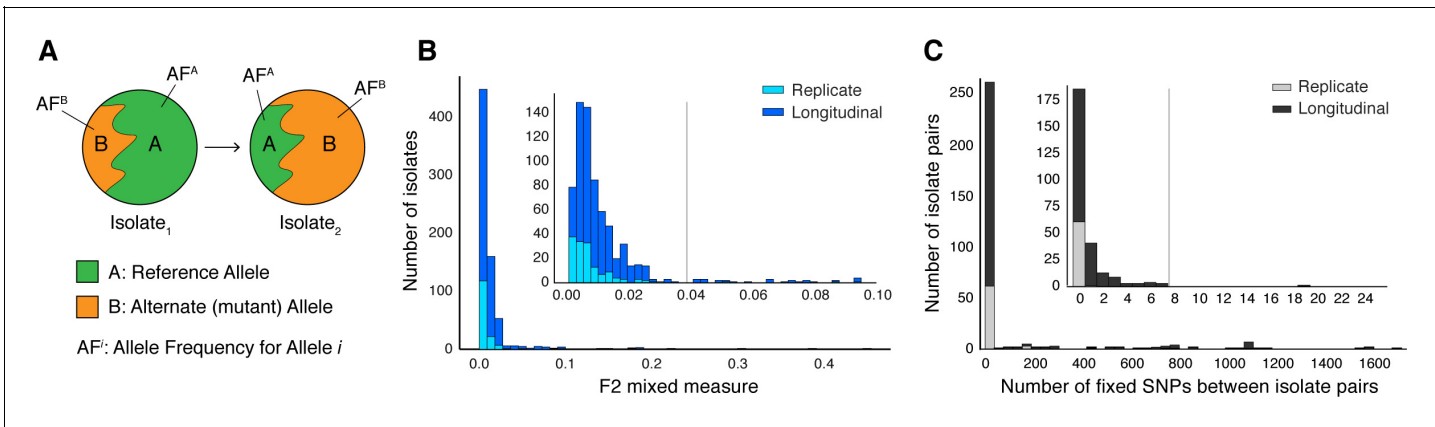

**Figure 1.** Selection of patients with longitudinal clonal infection. (**A**) Allele frequency change between paired isolates $(\Delta AF) = \left| AF_1^A - AF_2^A \right| = \left| AF_1^B - AF_2^B \right|$. (**B**) The F2 measure >0.04 (Materials and methods) was used to identify and exclude isolate pairs with evidence for mixed strain growth at any time point. (**C**) Replicate and longitudinal pairs with fixed SNP (fSNP) distance of >7 were excluded. For longitudinal isolates, fSNP >7 was assessed as consistent with Mtbc reinfection with a different strain.

The online version of this article includes the following figure supplement(s) for figure 1:

**Figure supplement 1.** Filtering out laboratory-contaminated samples and patients with mixed infections.

amplification and demonstrate that in addition to loci involved in the acquisition of antibiotic resistance, loci implicated in modulation of innate host-immunity appear to be under positive selection.

## Results

### Identifying clonal Mtbc populations in-host

Of the 307 patients with longitudinal samples collected (*Supplementary file 1–2*), 32 patients had evidence for isolate microbiological contamination at any time point (*Goig et al., 2020*) and were excluded. We found evidence for mixed infection with two or more Mtbc lineages (*Wyllie et al., 2018*) for 31 patients (*Figure 1B* and *Figure 1—figure supplement 1*); 44 patients had evidence for re-infection with a different Mtbc strain between the first and second time points, using a pairwise genetic distance >7 fixed SNPs (fSNPs) (Materials and methods, *Figure 1C* and *Figure 1—figure supplement 1*). Median fSNP distance for the 44 patients identified as reinfection was 708 (IQR 250–1086). The remaining 200 patients were accordingly identified as having persistent or relapsed clonal infection (*Supplementary file 3*). Isolates from these infections spanned five of the eight known Mtbc lineages (Figure 5A). We implemented WGS SNP calling filters to minimize the likelihood of false positive SNP calls and validated calls with simulation and PacBio long-read data. We required that no indels be present in any of the reads supporting any SNP call, dropped SNP calls in repetitive regions and enforced a read depth ≥25x and alternate allele depth of ≥5 reads. We estimated the false error rate of our analysis pipeline for detecting allele frequency changes between sampling times at ≤0.053 using a control dataset of 82 isolate pairs (162 total) that were in vitro technical or biological replicates (Materials and methods, Figure 4 and *Figure 1—figure supplement 1*).

### In-host pathogen dynamics in antibiotic resistance loci

Of the 200 patients with clonal infection, we had complete treatment data on 127 patients. Six of the 127 patients had isolates sampled <2 months apart, and the remaining 121 had an outcome at the second sampling consistent with delayed culture conversion, failure or relapse of their clonal infection, hitherto treatment failure for brevity. Treatment regimen details are provided in *Supplementary file 2*. Of the other 73 patients, 49 patients were sampled ≥2 months apart during treatment but regimen details or interruptions were not available, for these patients the outcome may have been either default or failure. The remaining 24 patients had inadequate treatment data to confirm treatment outcome. We conducted all analyses focused on antibiotic resistance loci on 200 patients with isolate date data and separately on the 121-patient subset with confirmed failure (the latter detailed in Appendix 3). For all 200 cases, the order of sampling was available, but for 195/200 (119/121 confirmed failure patients) we also had the exact dates of sampling which were required for some analyses.

Resistance mutations found at low frequencies in-host may indicate the impending development of clinical resistance (*Sun et al., 2012*; *Trauner et al., 2017*; *Zhang et al., 2016*). To investigate temporal dynamics related to antibiotic pressure, we identified non-synonymous and intergenic SNPs within a set of 36 predetermined resistance loci associated with antibiotic resistance (*Farhat et al., 2016*; *Farhat et al., 2013*; *Supplementary file 5*) that changed in allele frequency by ≥5% (*Sun et al., 2012*) and ensuring that support of the alternate allele was ≥5 reads at each time point (Materials and methods). We detected 1939 such SNPs across our sample of 200 patients (*Figure 2B*), 1774 were non-synonymous, 91 were intergenic, and 74 occurred within the *rrs* region (*Supplementary file 6*).

We searched for signs of selection by identifying clonal interference, or evidence of competition between strains with different drug resistance mutations (*Sun et al., 2012*; *Trauner et al., 2017*; *Zhang et al., 2016*). We characterized this in longitudinal isolates fulfilling three criteria: (i) isolates containing multiple resistance SNPs in the same gene within the same patient, (ii) at alternate allele frequencies that change in opposing directions over time, and (iii) the alternate (mutant) allele frequency was intermediate to high at ≥40% in at least one isolate (*Farhat et al., 2016*) for at least one of the co-occurring SNPs. This identified 11 cases of clonal interference (*Figure 2A* and *Figure 2—figure supplement 1*), demonstrating most often the fixation of a single allele in the second isolate from a mixture of multiple alleles at lower frequencies in the first isolate collected.

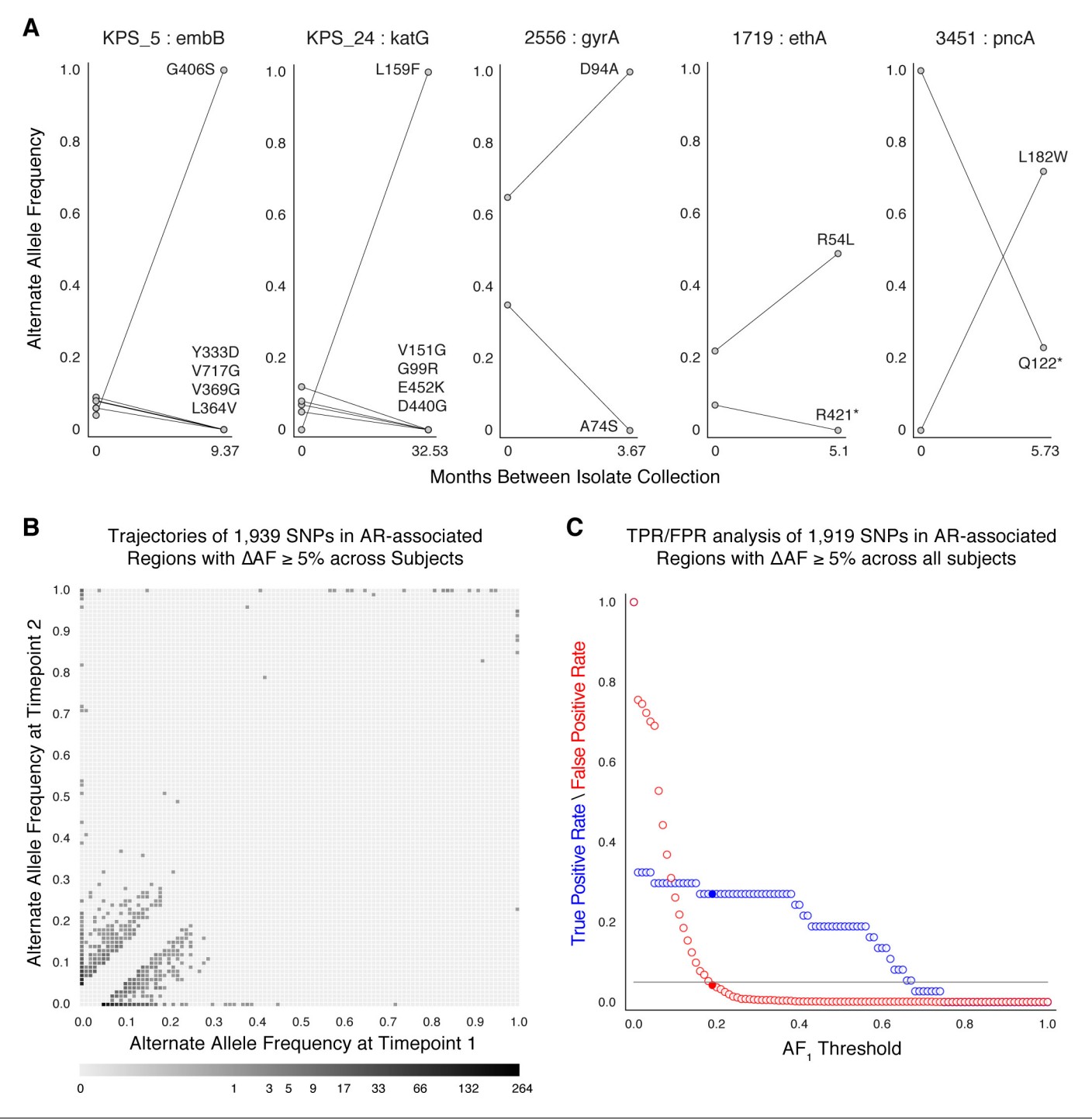

**Figure 2.** Allele frequency dynamics within antibiotic resistance loci. (A) The antibiotic resistance genes *embB*, *katG*, *gyrA*, *ethA*, and *pncA* demonstrate evidence for competing clones during infection (other examples found are displayed in ***Figure 2—figure supplement 1***). Each mutant allele is labeled with amino acid encoded by the reference allele, H37Rv codon position, and amino acid encoded by the mutant allele. (B) The allele frequency trajectories for SNPs that occur in patients over the course of infection can be used to study the prediction of further antibiotic resistance using the frequency of alternate alleles detected in the longitudinal isolates collected from patients. (C) Plot of true positive rate (TPR) and false positive rate (FPR) for detecting eventual fixation of a resistance allele ($AF_2 \geq 75\%$) as a function of initial allele frequency ($AF_1$).

The online version of this article includes the following figure supplement(s) for figure 2:

**Figure supplement 1.** Mutant allele trajectories consistent with clonal interference.

## Allele frequency >19% predicts subsequent fixation of resistance variants

We aimed to measure the lowest AR allele frequency that can accurately predict the fixation of resistance alleles later in time (*Dreyer et al., 2020*; *Sun et al., 2012*; *Zhang et al., 2016*). We examined all 1919 SNPs that varied by at least 5% in allele frequency (AF), and discarded 20 SNPs that were fixed at AF >75% in both isolates. We calculated the true positive rate (TPR) and false positive rate (FPR) for varying values of AF at the first time point ($AF_1 \in \{0, 1, 2, \cdots, 99, 100\}\%$ (*Figure 2C*, Materials and methods) allowing a maximum FPR of 5%. We found the optimal classification threshold to be $AF_1^* = 19\%$ with an associated sensitivity of 27.0% and a specificity of 95.8%. Of the total 37 alleles that became fixed at the second time point, 10 (from seven patients) had a frequency between 19% and 75% at the first time point, two were detected at the first time point but had AF <19%, and the remaining majority, or 25, were undetectable (i.e. had support of <5 reads) at the first time point. Taken together, we find a high turnover of low-frequency alleles in loci associated with antibiotic resistance but that mutant alleles in these loci that rise to a frequency of 19% are predicted to fix in-host with a sensitivity of 27.0% and specificity of 95.8%.

## Determinants of antibiotic resistance acquisition and microbiological treatment failure

We identified overall rates of resistance acquisition by focusing on AR SNPs with moderate to high $\Delta AF \geq 40\%$ given prior evidence of association between such SNPs and phenotypic resistance (*Farhat et al., 2016*).

Twenty-seven AR SNPs were acquired in the 178 patients with persistent or relapsed clonal infection $\geq 2$ months (*Figure 3B*). Among the set of 119 patients with confirmed failure and known isolate sampling date, 9% (11/119) of these patients acquired $\geq 1$ AR SNP. Of the 11, 9 patients received fewer than four effective drugs. We examined the relationship between pre-existing resistance and new AR acquisition. Pre-existing resistance was defined as $\geq 1$ fixed AR SNPs in the first isolate (*Farhat et al., 2016*) (Materials and methods). Two hundred fifty-nine pre-existing AR SNPs were identified with 41% (73/178) of failure patients harboring resistance to any drug at the first sampling (*Figure 3B*, *Supplementary file 7*). The majority of this resistance was MDR (multidrug resistance to at least isoniazid and a rifamycin), 64% (47/73) (*Figure 3C*). New resistance acquisition occurred mostly in patients with pre-existing resistance 20/27 (74%) ($OR = 5.28$, $P = 2.2 \times 10^{-4}$ Fisher's exact test) or pre-existing MDR ($OR = 3.85$, $P = 3.4 \times 10^{-3}$ Fisher's exact test). Among the set of 195/200 patients with clonal infection and sampling date, AR acquisition was more likely as the time between sampling increased with the OR of AR acquisition being 1.023 per 30day increment (95% $CI$ 1.002, 1.045, $P = 0.035$ Logistic Regression).

We also quantified genome-wide Mtbc diversity in-host among the patients with persistent or relapsed infection for $\geq 2$ months. We reasoned that if these patients are not on or not adherent to effective antibiotic treatment, their effective pathogen population size may be large and prone to more genetic drift or turnover of minority variants with and without selection (*Trauner et al., 2017*). We counted the number SNPs with an alternate allele frequency between 25% and 75% (Materials and methods) at each time point as a conservative estimate of the number of segregating sites in each population. We found this count to strongly correlate between the first and second time point (*Figure 3A*) suggesting that minor allele diversity is maintained in-host in patients without effective therapy (median T1=13.5 hSNPs, median T2=13.5 hSNPs, $r^2 = 0.426$, $P = 5.97 \times 10^{-23}$ Linear Regression).

## Genome-wide in-host diversity

Beyond antibiotic pressure, selective forces acting on the infecting Mtbc population in-host are largely unknown. To investigate this reliably across the entire Mtbc genome, we first examined the genome-wide allele frequency distribution for both technical replicates (in vitro technical or biological replicates, sample size m=62 after exclusions) and in-host longitudinal pairs (*Figure 4* and *Figure 1—figure supplement 1*). We detected five SNPs in *glpK* (with $\Delta AF \geq 25\%$) among five replicate pairs (mean $\Delta AF = 45\%$) consistent with an adaptive role for *glpK* mutations in vitro (*Pethe et al., 2010*; *Vargas and Farhat, 2020*) and accordingly excluded this gene from further analysis (Materials and methods). The genome-wide AF distribution in both replicate and

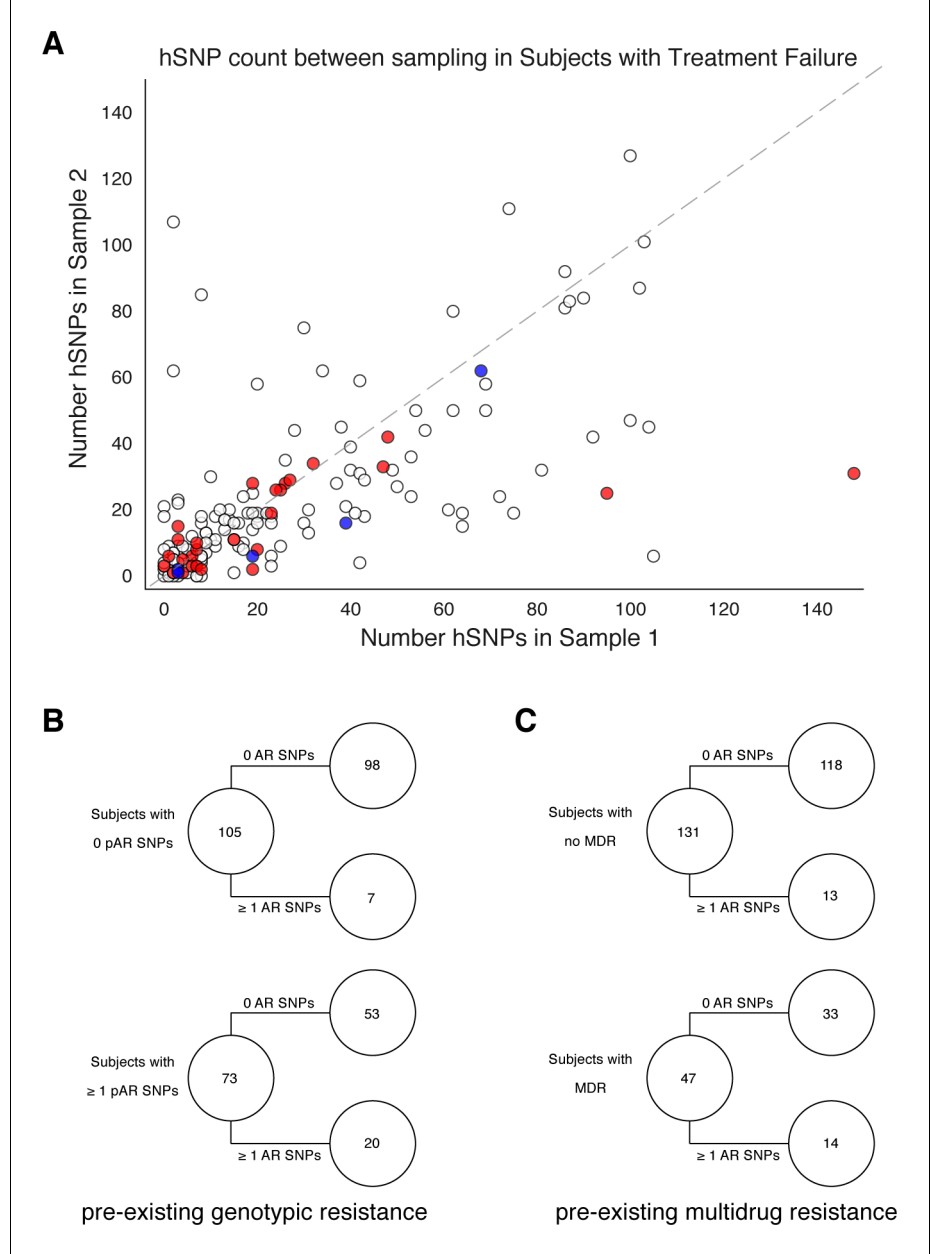

**Figure 3.** Pre-existing resistance is associated with resistance amplification. We called heterozygous SNPs (hSNP) in each isolate from a patient with clonal infection classified as failing treatment (N = 178). We defined hSNPs as a SNP called in an isolate with an alternate allele frequency between 25% and 75% (Materials and methods). (**A**) The number of hSNPs called in the second sample isolated vs the number of hSNPs called in the first sample isolated from each of 178 patients (median T1 = 13.5 hSNPs, median T2 = 13.5 hSNPs). The dashed line is y = x. Red denotes 27/178 patients who had an antibiotic resistance in-host SNP arise between sampling (median T1 = 15.0 hSNPs, median T2 = 11.0 hSNPs), blue denotes 5/178 patients who had a putative host-adaptive in-host SNP (Rv1944c, Rv0095c, *PPE18*, *PPE54*, *PPE60*) arise between sampling (median T1 = 19.0 hSNPs, median T2 = 6.0 hSNPs). (**B–C**) Among patients who fail treatment, (**B**) patients with pre-existing mutations that confer antibiotic resistance and (**C**) those that have pre-existing MDR are more likely to acquire antibiotic resistance mutations throughout the course of infection.

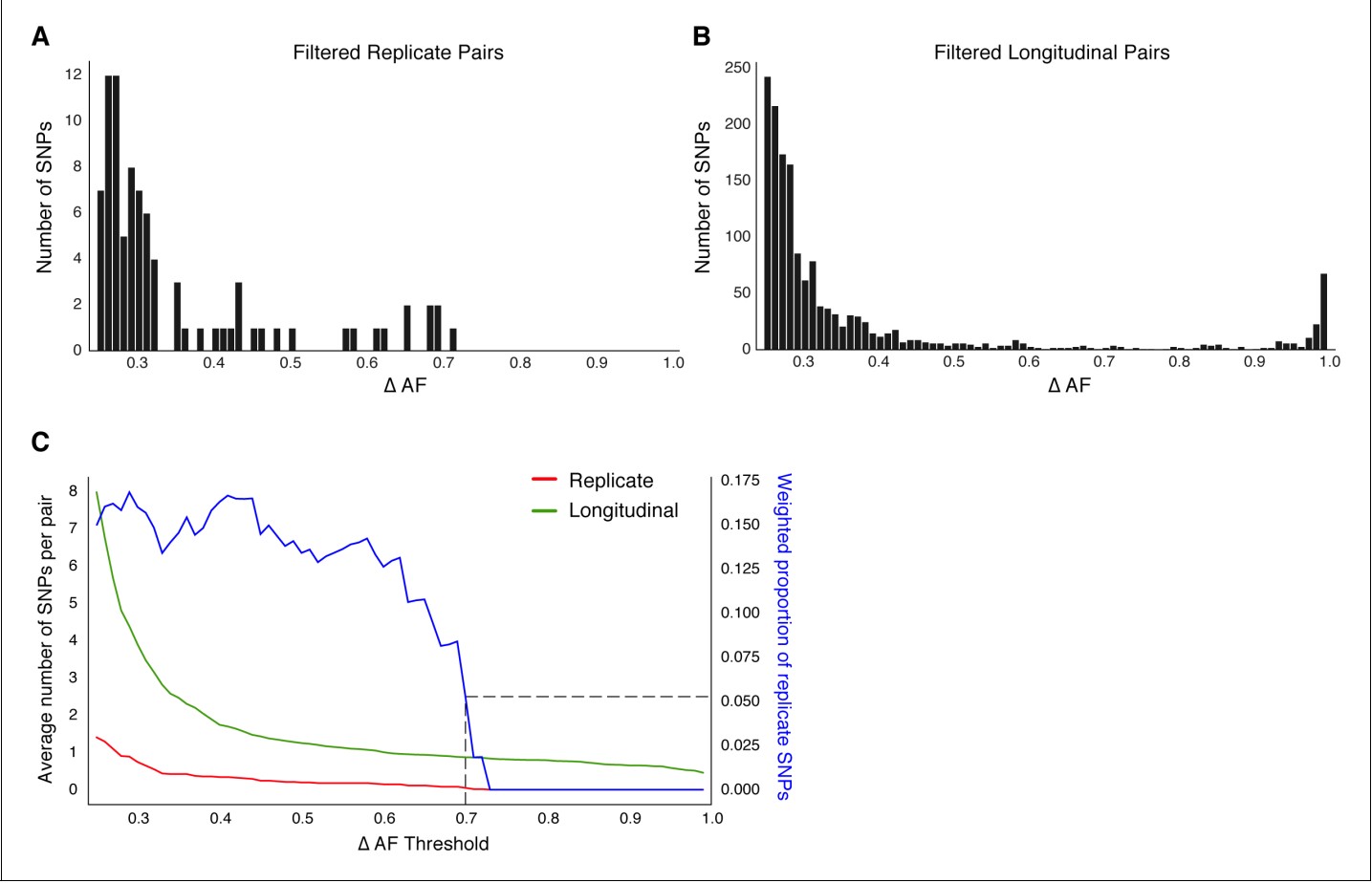

**Figure 4.** Replicate pairs reveal levels of biological noise associated with repeated sampling. (A, B) We analyzed the distribution of ΔAF for all SNPs detected across all replicate pairs ($m = 62$) and longitudinal pairs ($n = 200$) for SNPs where ΔAF ≥25%. (B) SNPs were detectable at lower levels of ΔAF for both types of isolate pairs, but SNPs with higher values of ΔAF were only found in longitudinal pairs. (C) To determine a ΔAF threshold for calling SNPs representative of changes in bacterial population composition in-host, we calculated the average number of SNPs per pair of isolates at different ΔAF thresholds for both replicate and longitudinal pairs. At a ΔAF threshold of 70% the number of SNPs between replicate pairs represents 5.27% of the SNPs detected amongst all replicate and longitudinal pairs, weighted by the number of pairs in each group.

longitudinal pairs demonstrated an abundance of SNPs with low ΔAF likely resulting from noise or technical factors. To clearly distinguish signal related to in-host factors from noise, we determined the ΔAF threshold above which SNPs/isolate-pair were rare among technical replicates that is, constituted 5% or less of total SNPs (*Figure 4*). We determined this ΔAF threshold to be 70% and selected 174 SNPs that developed in-host (in-host SNPs) among the 200 TB cases (*Figure 4C*, *Supplementary file 8*). Using archived MTBC isolates, we observe that changes in allele frequency are common among replicate isolates and changes in frequency of 70% are indicative of in-host evolution.

## Characteristics of mutations in-host

Of the 174 SNPs, 112 were non-synonymous, 42 synonymous, and 21 were intergenic (*Figure 5C*). The 153/174 coding SNPs were distributed across 127/3,886 genes and were observed in 71/200 patients (*Figure 5B* and *Figure 5D*). We analyzed the spectrum of mutations and found the GC > AT nucleotide transition to be the most common. The GC > AT transition is putatively due to oxidative damage including the deamination of cytosine/5-methyl-cytosine or the formation of 8-oxoguanine (*Dillon et al., 2015*; *Ford et al., 2011*). The transversion AT > TA was the least common substitution (*Figure 6A*). We expected the number of SNPs detected between longitudinal isolates to increase with time between isolate collection. Regressing the number of SNPs per patient on the

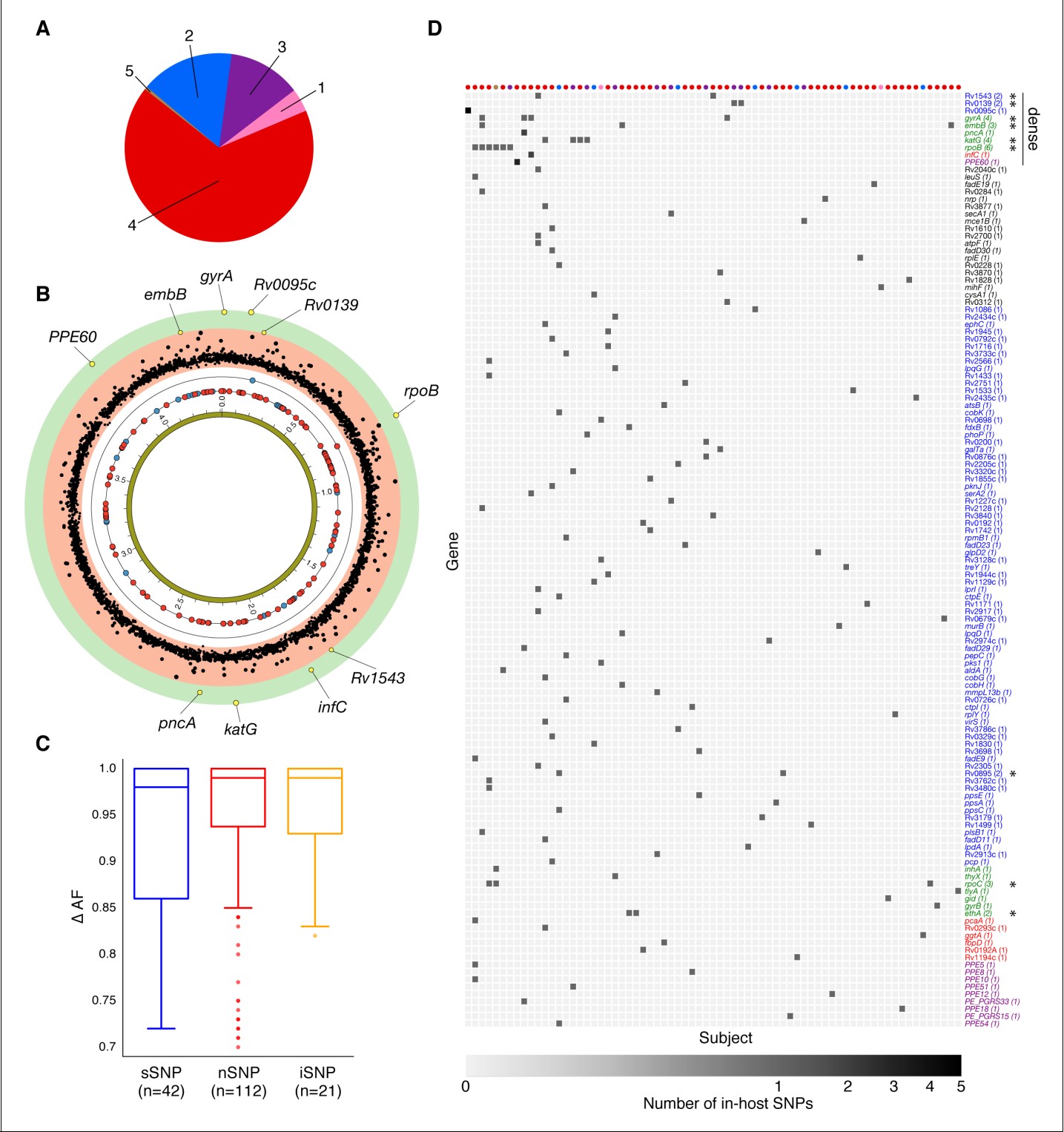

**Figure 5.** Genome-wide diversity in 200 clonal Mtbc infections. (A) Distribution of five major Mtbc lineages among the 200 clonal Mtbc infections. (B) Distribution of 153 in-host SNPs within coding regions among the 200 longitudinal isolate pairs across the 4.41 Mbp Mtbc genome (blue circles: synonymous, red circles: non-synonymous). Blue and red circles on the innermost black ring indicate the locations of SNPs detected in one patient; circles on the next ring represent SNPs detected in two patients. The $-\log_{10}$ (p-value) of the mutational density test (Materials and methods) by gene is plotted in the outermost, red and green, regions. Labeled yellow circles represent genes significant at the bonferroni-corrected cutoff ($\alpha = 0.05/3,886$). (C) Distribution of ΔAF by SNP type: sSNP: synonymous, nSNP: non-synonymous, iSNP: intergenic. (D) Heat-map of SNPs per gene (rows) and patient (columns). Colored circles across columns indicate the strain phylogenetic lineage (as represented in A). Gene names colored according to gene

*Figure 5 continued on next page*

*Figure 5 continued*

category (*Figure 6D*) with parentheses indicating the number of patients with an SNP in a given gene. *Indicates genes in which SNPs are detected within multiple patients.

The online version of this article includes the following figure supplement(s) for figure 5:

**Figure supplement 1.** In-host SNP detected in *PPE18*.
**Figure supplement 2.** In-host SNP detected in *PPE54*.
**Figure supplement 3.** In-host SNPs detected in *PPE60*.

timing between isolate collection (for 195 patients with isolate collection dates) (*Figure 6B*), we found SNPs to accumulate at an average rate of 0.56 SNPs per genome per year ($P = 7 \times 10^{-12}$) consistent with prior in vivo estimates (*Ford et al., 2011*; *Walker et al., 2013*).

## Simulations and PacBio sequencing demonstrate a low false-positive rate in repetitive regions

Several SNPs detected were in the GC-rich repetitive PE/PPE gene family (*Brennan and Delogu, 2002*). Variants called on these genes are commonly excluded from comparative genomic analyses (*Casali et al., 2016*; *Comas et al., 2010*; *Copin et al., 2016*; *Coscolla et al., 2015*) due to the limitations of short-read sequencing data and the possibility of making spurious variant calls; however, the rates at which these false calls occur has not been evaluated. We reasoned that our stringent filtering criteria, quality of sequencing data and depth of coverage allowed us to reliably detect variants in these regions of the genome, with the potential to uncover variation in these understudied regions of the genome.

We took several approaches to test the rate of false-positives for the single base-pair mutations observed in our analysis (Materials and methods). First, we introduced the mutant alleles observed in-host (*Supplementary file 10*) into a set of Mtbc reference genomes belonging to different lineages and simulated short read sequencing data from these modified genomes (*Appendix 1—figure 1*). We then used our variant calling pipeline to call bases from this simulated data. We observed a high recall rate of the introduced mutant alleles and a very low number of false positive base calls (zero in most cases) within the loci containing modified alleles (*Appendix 1—figure 2*). Second, we assessed the congruence in variant calls between short-read Illumina data and long-read PacBio data for a set of isolates that underwent sequencing with both technologies (Materials and methods). Unlike Illumina generated reads, PacBio reads are much longer and have randomly distributed error profiles (*Rhoads and Au, 2015*). With high coverage, PacBio sequencing can reliably reconstruct full microbial genomes and identify SNPs in repetitive regions. The comparison with PacBio assemblies confirmed empirically a low rate of false positive base calls in genomic regions where we observed in-host SNPs (Materials and methods). Third, we confirmed the five phylogenetically convergent in-host SNPs in PPE genes *PPE18*, *PPE54*, and *PPE60* (see below) through manual inspection of the read alignment (*Figure 5—figure supplements 1–3*).

## Antibiotic resistance and PE/PPE genes vary while antigens remain conserved

To understand how different classes of proteins evolve in-host, we separated Mtbc genes into five non-redundant categories (Materials and methods). The vast majority of genes in each category did not vary within patients (*Figure 6C*). Antibiotic resistance genes were on average the most diverse category while Essential genes varied the least (*Figure 6D*). Antigen genes appeared to be as conserved as were both Essential ($P = 0.49$ Mann-Whitney U-test) and Non-Essential genes ($P = 0.45$ Mann-Whitney U-test) while PE/PPE genes showed higher levels of nucleotide diversity than both Essential ($P = 0.022$ Mann-Whitney U-test) and Non-Essential genes ($P = 0.013$ Mann-Whitney U-test) (*Figure 6D*).

## PE/PPE variation is independent of T-cell recognition

To test whether variation in Antigen or PE/PPE genes occurred in response to T-cell recognition, we separated each gene in these categories into (CD4$^+$ and CD8$^+$ T-cell) epitope and non-epitope concatenates and recalculated nucleotide diversity for these concatenates (*Figure 6E–H*). For both

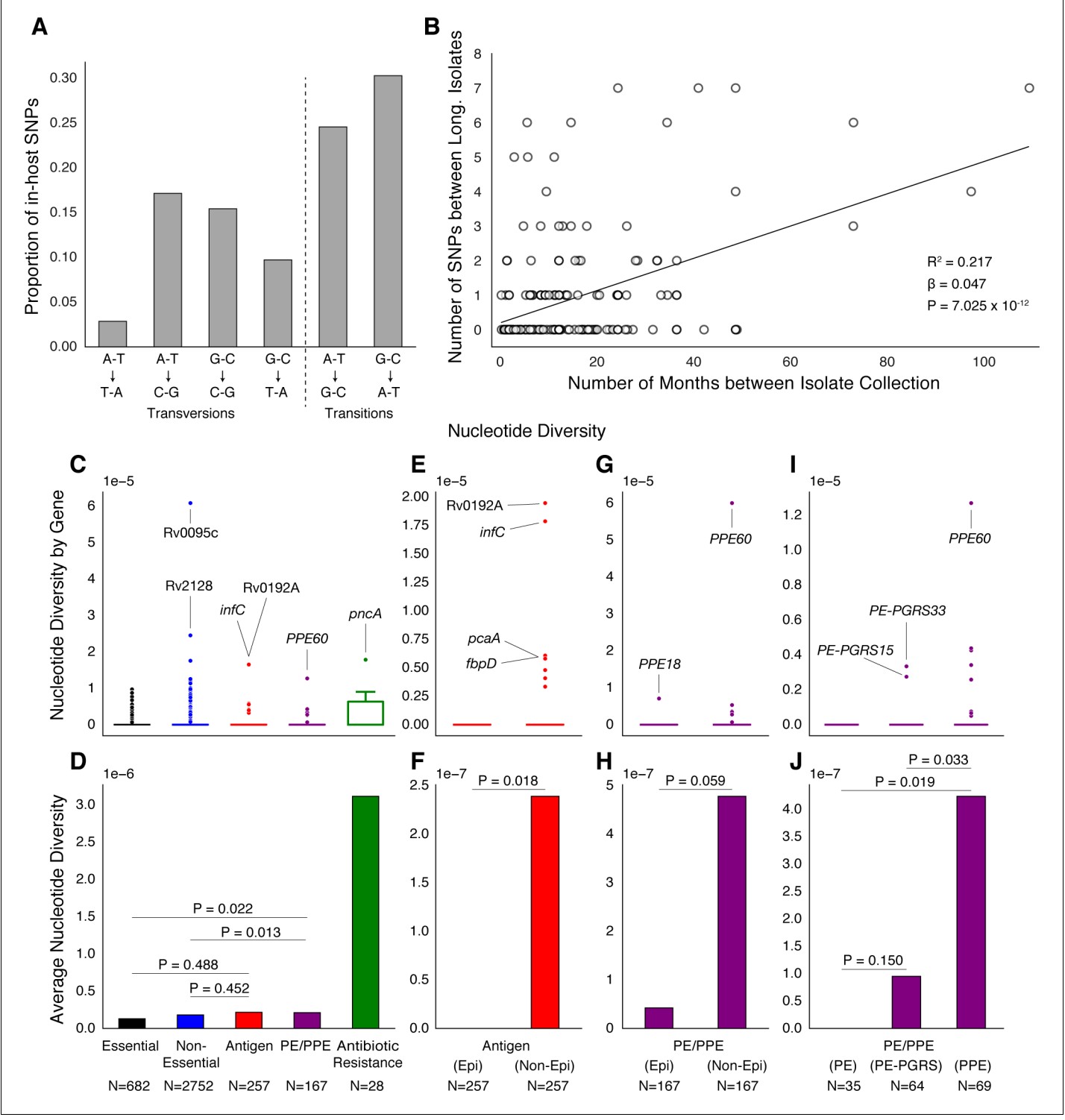

**Figure 6.** PE/PPE genes vary considerably within host while putative antigens remain conserved. (**A**) Mutational spectrum of in-host SNPs. (**B**) In-host SNP counts vs. time between isolate collection (195/200 patients with dates shown, *W [*Walker et al., 2013*] isolates only had year of collection). (**C**) Boxplots of nucleotide diversity by gene within each of five non-redundant categories (see text; $n$ = number of genes). (**D**) Average nucleotide diversity across genes by category. Nucleotide diversity in epitope and non-epitope region (Materials and methods) of each gene in the Antigen (**E**, **F**) and PE/PPE (**G**, **H**) gene categories. (**I**, **J**) PE/PPE genes separated into three non-redundant categories: PE, PE-PGRS, and PPE. (**J**) The average nucleotide diversity by category. (**I**) Box plot of nucleotide diversity by gene.

The online version of this article includes the following figure supplement(s) for figure 6:

*Figure 6 continued on next page*

Figure 6 continued

**Figure supplement 1.** Basic characteristics of epitopes used in analysis.
**Figure supplement 2.** Most T cell epitopes remain conserved in-host during active TB disease.

Antigen and PE/PPE genes (*Figure 6F* and *Figure 6H*), epitope concatenates were less diverse than non-epitope concatenates ($P = 0.018$ and $P = 0.059$, respectively, Mann-Whitney U-test). Only one in-host SNP was detected within an epitope-encoding region in the gene *PPE18* (*Figure 6G* and *Figure 7—figure supplement 2*, *Supplementary file 12*). This suggests that T-cell recognition does not drive diversity in these regions. Looking within the three PE/PPE subfamilies (*Figure 6I–J*; *Brennan, 2017*), the PPE genes appeared more diverse in-host than PE genes and PE-PGRS genes ($P = 0.019$ and $P = 0.033$ respectively, Mann-Whitney U-test).

## Identifying candidate pathoadaptive loci from genome-wide variation

To identify genes involved in pathogen adaptation (*Lieberman et al., 2011*; *Marvig et al., 2015*), we applied a test of mutational density (*Farhat et al., 2014*; Materials and methods) by pooling variation across all 200 pairs of genomes and identifying those genes with more mutations than expected under a neutral model of evolution where variants are Poisson distributed across the genome (*Farhat et al., 2014*; *Figure 5B*, *Supplementary file 13*, Materials and methods). We also searched for evidence of convergent evolution, that is, genes or pathways where in-host SNPs developed in $\geq 2$ patients (*Supplementary file 14*, *Supplementary file 17*). Seven known antibiotic resistance genes (*Didelot et al., 2016*; *Farhat et al., 2013*) had significant mutational density ($\alpha = 0.05$, Bonferroni correction) or were convergent across patients: *rpoB*, *gyrA*, *katG*, *rpoC*, *embB*, *ethA* and *pncA* (mutated in six, four, four, three, three, two, and one patient, respectively) (*Figure 5B* and *Figure 5D*). Single in-host SNPs occurred in eight additional known resistance loci including three intergenic regions, and in *prpR,* a gene recently implicated with drug tolerance (*Hicks et al., 2018*; *Supplementary file 8*).

Three genes with unknown function: Rv0139, Rv0895, and Rv1543 were convergent in two patients each, two of which (Rv0139, Rv1543) had significant mutational density ($p<2\times10^{-5}$) and; three additional genes including *PPE60* displayed significant mutational density ($p<2\times10^{-5}$) (*Figure 5B*, *Supplementary file 13*). We found evidence for convergence in six pathways not known to result in antibiotic resistance. These pathways are involved with biotin biosynthesis (*fadD23*, *fadD29*, and *fadD30*), ribosomal large subunit proteins (*rpmB1*, *rplE*, and *rplY*), glycerolipid and glycerophospholipid metabolism (*aldA* and Rv2974c), ESAT-6 protein secretion (*eccCa1* and *eccD1*), coenzyme B12/cobalamin synthesis (*cobH* and *cobK*) and the uncharacterized pathway CBSS-164757.7.peg.5020 (*fdxB* and *PPE18*) (*Supplementary file 17*).

## In-host mutations display phylogenetic convergence across multiple global lineages

We reasoned that pathoadaptive mutations observed to sweep to fixation in-host and not compromise pathogen transmissibility are likely to arise independently within other patients and in separate geographic regions in a convergent manner (*Farhat et al., 2013*). We screened a geographically diverse set of 20,352 sequenced clinical isolates belonging to global lineages 1–6 for mutations observed within host in which the alternate (mutant) allele swept over the course of sampling (141/174 in-host SNPs, Materials and methods, *Supplementary file 8*, *Supplementary file 18*). Conservatively, a mutation was characterized as phylogenetically convergent if it was present in isolates from three or more global lineages but not fixed in any lineage (Materials and methods). We identified 26/141 in-host SNPs as phylogenetically convergent in our global sample of isolates (*Supplementary file 19*). *Figure 7* and *Figure 7—figure supplements 1–2* display the distribution of convergent alleles across the 20,353 isolates using t-Distributed Stochastic Neighbor Embedding (t-SNE) of the pairwise genetic distance matrix (Materials and methods). The convergent alleles included the PPE genes *PPE18* (1 site), *PPE54* (1 site) and *PPE60* (3 sites), as well as Rv0095c (2 sites) and Rv1944c (1 site) both conserved proteins of unknown function. In addition to several SNPs in loci associated with antibiotic resistance, *gyrB* (1 site), *gyrA* (2 sites), *rpoB* (4 sites), *rpoC* (3 sites), *inhA* (1 site), *embB* (3 sites), and *gid* (1 site).

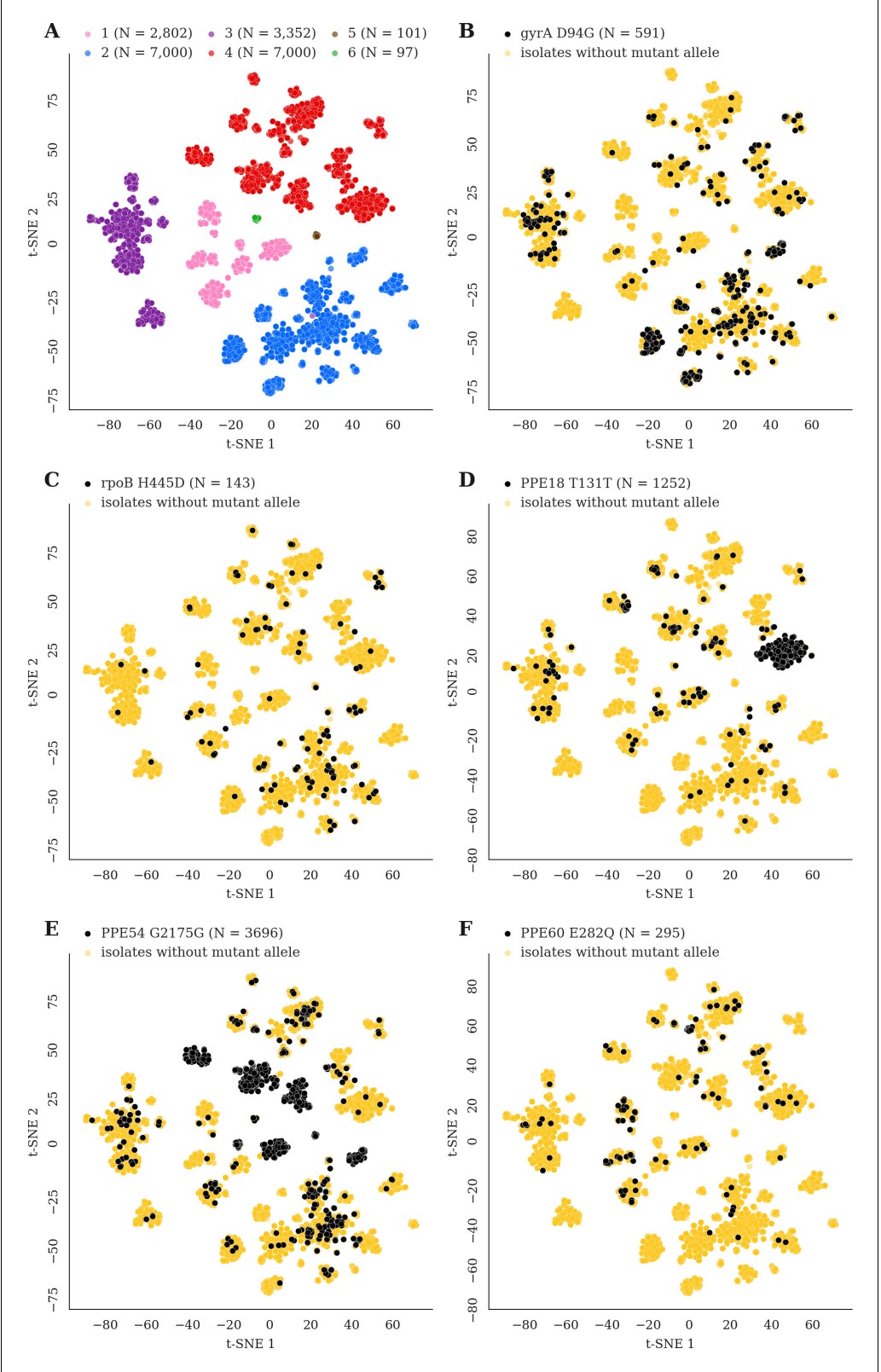

**Figure 7.** Mutations acquired in-host are phylogenetically convergent. We constructed t-SNE plots from a pairwise SNP distance matrix for our global sample of 20,352 clinical isolates and 128,898 SNP sites (Materials and methods). (A) Labeling isolates by global lineage revealed that isolates cluster according to genetic similarity. Next, we labeled isolates by whether they carried a mutant allele that was also detected in-host. (B–F) Mutations in *gyrA*, *rpoB*, *PPE18*, *PPE54*, and *PPE60* were detected in-host (*Supplementary file 8*), occur in a global collection of isolates (*Supplementary file 18*)
*Figure 7 continued on next page*

*Figure 7 continued*

and are scattered across the tSNE plots, indicating that they belong to genetically different clusters of isolates (*Supplementary file 19*). Furthermore, all mutations with a signal of phylogenetic convergence were detected in isolates belonging to different clusters, confirming that theses mutations must have arisen independently in different genetic backgrounds (*Figure 7—figure supplements 1–2*). Each plot is labeled with the gene name each mutation occurs within, amino acid encoded by the reference allele, H37Rv codon position, and amino acid encoded by the mutant allele. N = number of isolates with mutant allele.

The online version of this article includes the following figure supplement(s) for figure 7:

**Figure supplement 1.** Mutations acquired in-host are phylogenetically convergent.

**Figure supplement 2.** Mutations acquired in-host are phylogenetically convergent.

## Discussion

In our Mtbc populations sequenced from active TB patients enriched for negative treatment outcomes, we find a wealth of dynamics in genetic loci associated with antibiotic resistance, including a high turnover of minor variants. Known factors that determine treatment outcome are complex and include severity of lung disease, cavitation and adherence to treatment among others (*Imperial et al., 2018*). Additionally, resistance acquisition in the course of one infection is comparatively rare in most pathogenic bacteria (*Llewelyn et al., 2017*). Here, we observe that 9% of patients with confirmed delayed culture conversion, failure and relapse amplify resistance over time. Our findings of a higher rate of resistance acquisition in patients with MDR at the outset and with time between sampling, emphasize the importance of appropriately tailoring treatment regimens as well as close surveillance for microbiological clearance and resistance acquisition by phenotypic or genotypic means. The observed high rate of resistance acquisition also emphasizes Mtbc's biological adaptability and the long duration of drug pressure in vivo. In addition to clonal acquisition of resistance, we find that sequencing revealed a substantial proportion of mixed infection or reinfection (28% of samples collected $\geq$2 months apart). This high percentage suggests that patient treatment and control of disease transmission can be better guided if pathogen sequencing is routinely performed for cases with persistent positive cultures especially in high TB prevalence settings where reinfection is more likely. Reinfection can also introduce strains with a different antibiotic susceptibility profile requiring adjustment in the treatment regimen.

While prior studies have investigated the lowest resistance allele frequencies that can be detected in clinical sputum samples (*Dreyer et al., 2020*; *Trauner et al., 2017*), there is little information on the clinical relevance of these low frequency variants. We provide a proof-of-concept analysis that minor AR alleles, occurring at a frequency 19%, can predict fixation of the variant with a specificity >95% of mutations in-host, although we find the sensitivity of this threshold to be low. The low sensitivity is because the majority of alleles that sweep to fixation are actually not detectable at all at the first time point, suggesting that more frequent sampling may be needed. In the future, higher depth and more frequent sequencing can elucidate more clearly the role of minor AR allele detection in clinical management of TB treatment.

Various sources of noise contribute to allele frequency changes over time and challenge inference on bacterial composition in vivo. Here, we determined an appropriate threshold for identifying mutations in-host using average depth Mtbc WGS from cultured isolates and demonstrate the importance of including technical replicate WGS. While culturing sputa in vitro enriches Mtbc DNA for WGS it also creates experimental noise (*Vargas and Farhat, 2020*) and can purge some of the genetic diversity present in the sputum sample (*Nimmo et al., 2019*). The refinement of methods for DNA extraction directly from sputum (*Votintseva et al., 2017*), may allow the calling of relevant changes in allele frequencies at lower thresholds in future work. This would permit the unbiased study of loci that may be under frequency-dependent selection, where changes in allele frequencies would unlikely change by as much as 70% as we used here.

We detected 174 alleles rising to near fixation in-host across our sample of 200 patients. The observed distribution of variants including the high rate of non-synonymous substitutions and the predominance of GC > AT variants are consistent with the hypotheses of purifying pressure on synonymous variants and oxidative DNA damage, respectively (*Ford et al., 2011*; *Namouchi et al., 2012*) in Mtbc. This consistency adds validity to our variant calling approach. Overall, the observed diversity spared the CD4$^+$ and CD8$^+$ T cell epitope encoding regions of the genome providing

further evidence that host adaptive immunity does not drive directional selection in Mtbc genomes now at short-time scales (*Comas et al., 2010*; *Copin et al., 2014*; *Coscolla et al., 2015*). Diversity was concentrated in antibiotic resistance regions and strikingly also in PE/PPE genes (*Figure 6D*; *Phelan et al., 2016*). Although previous studies have generally avoided reporting short-read variant calls in PE/PPE regions, we demonstrate using read simulation, visualization of Illumina read alignments and comparison with long-read sequencing data the accuracy of the SNPs captured in our study. We found PPE genes to be more diverse in-host than PE genes and detected a signal of positive selection acting on three genes belonging to the PPE sub-family (*Figure 7*). This indicates that PPE genes may be play an important role in the process of host-adaptation.

In addition to identifying in-host variation in 12 loci known to be involved in the acquisition of antibiotic resistance, we identified six genes and six pathways displaying diversity in-host and not known to be associated with antibiotic resistance (*Supplementary file 8*, *Supplementary file 13–14*, *Supplementary file 17*). For a subset, we demonstrate similar diversity has arisen independently in separate hosts and in strains with different genetic backgrounds suggesting positive selection (*Figure 7*). Evidence of directional selection in Mtbc genomes have thus far been largely restricted to adaptation to antibiotic treatment (*Brites and Gagneux, 2015*; *Didelot et al., 2016*; *Trauner et al., 2017*). The novel pathways showing in-host convergence may be important for interactions between host and pathogen arising from either metabolic or immune pressure. Mtbc is one of a few types of bacteria that possess the capacity for de novo coenzyme B12/cobalamin synthesis, and this pathway has been implicated in Mtbc survival in-host and Mtbc growth (*Rowley and Kendall, 2019*). We identified four genetic variants that developed in three separate patients and in three consecutive genes from the same locus *cobG*, intergenic *cobG-cobH*, *cobH* and *cobK* (Rv2064-Rv2067). This observation contributes to mounting evidence on the importance of this pathway for in vivo Mtbc survival and may have implications for drug development (*Gopinath et al., 2013*; *Minias et al., 2018*). Biotin biosynthesis is also relatively unique to mycobacteria and plays an important role in Mtbc growth, infection and host survival during latency (*Salaemae et al., 2011*). The other identified pathways include ESAT-6 protein secretion known to play a role in the modulation of host immune response by disrupting the phagosomal membrane (*Clemmensen et al., 2017*).

The loci found to be phylogenetically convergent and not known to be associated with antibiotic resistance, include the genes Rv0095c, *PPE18*, *PPE54*, and *PPE60*. Consistent with the idea that positive selection is acting on alleles within these loci, we observe a reduction in diversity at the second time point for the patients in which drug-resistant alleles sweep to fixation and in which putative host-pathogen alleles sweep to fixation (*Figure 3A*). Although of unknown function, Rv0095c (SNP A85V) was recently associated with transmission success of an Mtbc cluster in Peru (*Dixit et al., 2019*). Both *PPE18* and *PPE60* have been shown to interact with toll-like receptor 2 (TLR2) (*Nair et al., 2009*; *Su et al., 2018*). *PPE18* was the only gene to encode an epitope containing a SNP in-host; mutations in the epitope-encoding regions of this gene have previously been described in a set of geographically separated clinical isolates (*Hebert et al., 2007*). Furthermore, *PPE18* codes for one of the antigens used in the construction of the M72/AS01E vaccine candidate (*Tait et al., 2019*). Our results demonstrating that *PPE18* is under positive selection in the MTBC may have implications for the efficacy of this vaccine against genetically diverse Mtbc strains. *PPE54* has been implicated in Mtbc's ability to arrest macrophage phagosomal maturation (phagosome-lysosome fusion) and thought to be vital for intracellular persistence (*Brodin et al., 2010*). The mechanism by which *PPE54* accomplishes this is unknown, but Mtbc modification of phagosomal function is thought to be TLR2/TLR4-dependent (*Podinovskaia et al., 2013*).

Mtbc is known to disrupt numerous *innate* immune mechanisms including phagosome maturation, apoptosis, autophagy as well as inhibition of MHC II expression through prolonged engagement with innate sensor toll-like receptor 2 (TLR2) among others (*Ernst, 2018*). SNPs in human genes involved with innate-immune pathways have been implicated in-host susceptibility to TB (*Azad et al., 2012*; *Kleinnijenhuis et al., 2011*; *Tientcheu et al., 2017*). Specifically, SNPs in TLR2 (thought to be the most important TLR in Mtbc recognition) (*Tientcheu et al., 2017*) and TLR4 have been associated with susceptibility to TB disease (*Azad et al., 2012*; *Kleinnijenhuis et al., 2011*). Taken together, these observations and our results are consistent with ongoing co-evolution between humans and Mtbc with evidence for reciprocal adaptive changes, leaving a signature of selection in both humans and Mtbc populations (*Brites and Gagneux, 2015*). Most co-evolution between Mtbc and humans, the main reciprocal adaptations between host and pathogen are

thought to have occurred long ago and as a result of long-term host-pathogen interactions (*Azad et al., 2012*; *Brites and Gagneux, 2015*). Here, we observe these dynamics over the short evolutionary timescale of a single infection which has important implications for vaccine development (*Brennan, 2017*).

## Materials and methods

### Sequence data

#### Longitudinal isolate pairs

This study included data for 614 clinical isolates of *M. tuberculosis* that were sampled from the sputum of 307 patients resulting in n = 307 longitudinal pairs. The sequencing data for 456 publicly available isolates was downloaded from Genbank (RRID:SCR_002760; *Benson et al., 2009*), sequenced using Illumina chemistry to generate paired-end reads and came from previously published studies (T [*Trauner et al., 2017*], C [*Casali et al., 2016*], W [*Walker et al., 2013*], B [*Bryant et al., 2013*], G [*Guerra-Assunção et al., 2015*], X [*Xu et al., 2018*], H [*Witney et al., 2017*], P [*Farhat et al., 2019*]; *Figure 1—figure supplement 1*, *Supplementary file 1–2*). We aggregated treatment from the source studies and added this metadata to *Supplementary file 2* for each longitudinal isolate. We include columns that indicate the timing of sampling of Mtbc relative to treatment, the treatment regimen administered and final patient outcome (and relevant details). Patient outcomes are defined as follows: *Delayed culture conversion* (sputum culture positive at baseline and 2 months treatment initiation with genomic analysis consistent with clonal infection), *Failure or Relapse* (sputum culture positive at baseline and 4.5 months treatment initiation with genomic analysis consistent with clonal infection), *Failure or Relapse or Default* (sputum culture positive at interval of 4.5 months with genomic analysis consistent with clonal infection, only partial treatment data is available) or N/A if date data was of low resolution, not available or no treatment data was available. We also determined *Reinfection* and *Mixed infection* based on the genomic analysis.

#### Replicate isolate pairs

This study included three types of replicate isolate pairs. (S2 - Sequenced Twice) DNA pooled from a single Mtbc clinical isolate that had undergone in vitro expansion was sequenced in separate runs on an Illumina sequencing machine (m = 5). (C2 – Cultured and Sequenced Twice) Mtbc was cultured from a single frozen clinical sample at separate time points, then sequenced on an Illumina sequencing machine after DNA extraction from culture (m = 73). (P3) Three sputum samples were obtained from a single patient within a 24-hr period (*Trauner et al., 2017*), cultured separately, underwent DNA extraction and then sequencing on an Illumina sequencing machine. For the purposes of this study, we compared these three isolates pairwise (m = 3).

#### Global sequence data

We downloaded raw sequence data for 33,873 clinical isolates from the public domain (*Benson et al., 2009*). Isolates had to meet the following quality control measures for inclusion in our study: (i) at least 90% of the reads had to be taxonomically classified as belonging to the *M. tuberculosis* complex after running the trimmed FASTQ files through Kraken (*Wood and Salzberg, 2014*) and (ii) at least 95% of bases had to have coverage of at least 10x after mapping the processed reads to the H37Rv Reference Genome.

### Epitope collection and analysis

CD4[+] T and CD8[+] T cell epitope sequences were downloaded from the Immune Epitope Database (RRID: SCR_006604) (*Vita et al., 2015*) on May 23rd, 2018 according to criteria described previously (*Coscolla et al., 2015*) [linear peptides, *M. tuberculosis* complex (ID:77643, Mycobacterium complex), positive assays only, T cell assays, any MHC restriction, host: humans, any diseases, any reference type] yielding a set of 2031 epitope sequences (*Supplementary file 11*). We mapped each epitope sequence to the genes encoded by the H37Rv Reference Genome (*Cole et al., 1998*) using BlastP with an e-value cutoff of 0.01 (*Figure 6—figure supplement 1A*). We retained only epitope sequences that mapped to at least one region in H37Rv (due to sequence homology, some epitopes mapped to multiple regions) and whose BlastP peptide start/end coordinates matched those

specified in IEDB (n = 1949,949 representing 1505 separate epitope entries in IEDB). We then filtered out any epitopes occurring in Mobile Genetic Elements which resulted in a final set of 1875 epitope sequences, representing 348 genes (antigens) used for downstream analysis. The distribution of peptide lengths for this final set of epitopes is given in *Figure 6—figure supplement 1B*. Since many of these epitope sequences overlap, we constructed non-redundant epitope concatenate sequences for each antigen (n = 348) gene (*Comas et al., 2010*; *Coscolla et al., 2015*; *Stucki et al., 2016*). The regions of each antigen not encoding an epitope were concatenated into a non-epitope sequence for that gene.

## Gene sets

Every gene on H37Rv was classified into one of six non-redundant gene categories according to the following criteria: (i) genes identified as belonging to the PE/PPE family of genes unique to pathogenic mycobacteria, though to influence immunopathogenicity and characterized by conserved proline-glutamate (PE) and proline-proline-glutamate (PPE) motifs at the N protein termini (*Brennan and Delogu, 2002*; *Comas et al., 2010*; *Phelan et al., 2016*) were classified as *PE/PPE* (n = 167), (ii) genes flagged as being associated with antibiotic resistance (*Farhat et al., 2013*) were classified into the *Antibiotic Resistance* category (n = 28), (iii) genes encoding a CD4$^+$ or CD8$^+$ T-cell epitope (*Comas et al., 2010*; *Coscolla et al., 2015*; but not already classified as a PE/PPE or Antibiotic Resistance gene) were classified as an *Antigen* (n = 257), (iv) genes required for growth in vitro (*Sassetti et al., 2003*) and in vivo (*Sassetti and Rubin, 2003*) and not already placed into a category above were classified as *Essential* genes (n = 682), (v) genes flagged as transposases, integrases, phages, or insertion sequences were classified as *Mobile Genetic Elements* (*Comas et al., 2010*) (n = 108), (vi) any remaining genes not already classified above were placed into the *Non-Essential* category (n = 2752) (*Supplementary file 4*).

## Illumina sequencing FastQ processing and mapping to H37Rv

The raw sequence reads from all sequenced isolates were trimmed with Prinseq (*Schmieder and Edwards, 2011*) (settings: `-min_qual_mean 20`) (version 0.20.4) then aligned to the H37Rv Reference Genome (Genbank accession: NC_000962) with the BWA mem (*Li and Durbin, 2009*) algorithm (settings: -M) (version 0.7.15). The resulting SAM files were then sorted (settings: `SORT_ORDER = coordinate`), converted to BAM format and processed for duplicate removal with Picard (http://broadinstitute.github.io/picard/) (version 2.8.0) (settings: `REMOVE_DUPLICATES = true, ASSUME_SORT_ORDER = coordinate`). The processed BAM files were then indexed with Samtools (*Li et al., 2009*). We used Pilon (*Walker et al., 2014*) on the resulting BAM files to call bases for all reference positions corresponding to H37Rv from pileup (settings: `-variant`).

## Empirical score for difficult-to-call regions

We extracted DNA from 15 Mtbc isolates (*Epperson and Strong, 2020*), for which we had Illumina sequencing reads, to undergo PacBio sequencing (Appendix 2). Together, with public PacBio and Illumina sequencing data (*Chiner-Oms et al., 2019*), we compiled 31 pairs of sequencing reads for comparison of variant calling between PacBio long-reads and Illumina short-reads (*Supplementary file 20*). Using the 31 isolates for which both Illumina and a complete PacBio assembly were available (Appendix 2), we evaluated the empirical base-pair recall (EBR) of all base-pair positions of the H37rv reference genome (Marin et al., in preparation). For each sample, the alignments (from minimap2) of each high confidence genome assembly to the H37Rv genome were used to infer the true nucleotide identity of each base pair position. To calculate the empirical base-pair recall, we calculated what % of the time our Illumina based variant calling pipeline, across 31 samples, confidently called the true nucleotide identity at a given genomic position. If Pilon variant calls did not produce a confident base call (*Pass*) for the position, it did not count as a correct base call. This yields a metric ranging from 0.0 to 1.0 for the consistency by which each base-pair is both confidently and correctly sequenced by our Illumina WGS-based variant calling pipeline for each position on the H37Rv reference genome. An H37Rv position with an EBR score of x% indicates that the base calls made from Illumina sequencing and mapping to H37Rv agreed with the base calls made from the PacBio de novo assemblies (Appendix 2) in x% of the Illumina-PacBio pairs. We

masked difficult-to-call regions by dropping H37Rv positions with an EBR score below 0.8 as part of our variant calling procedure.

## Variant calling

### Single-nucleotide polymorphism (SNP) calling

To prune out low-quality base calls that may have arisen due to sequencing or mapping error, we dropped any base calls that did not meet any of the following criteria (*Copin et al., 2016*): (i) the call was flagged as either *Pass* or *Ambiguous* by Pilon, (ii) the reads aligning to that position supported at most two alleles (ensuring that 1 allele matched the reference allele if there were 2), (iii) the mean base quality at the locus was > 20, (iv) the mean mapping quality at the locus was > 30, (v) none of the reads aligning to the locus supported an insertion or deletion, (vi) a minimum coverage of 25 reads at the position, (vii) the EBR score for the position $\geq 0.80$, and (viii) the position is not located in a mobile genetic element region of the reference genome. We then used the Pilon-generated (*Walker et al., 2014*) VCF files to calculate the frequencies for both the reference and alternate alleles, using the *INFO.QP* field (which gives the proportion of reads supporting each base weighted by the base and mapping quality of the reads, *BQ* and *MQ* respectively, at the specific position) to determine the proportion of reads supporting each base for each locus of interest.

### Additional SNP filtering for isolate pairs

To call SNPs (and corresponding changes in allele frequencies) between pairs of isolates (Replicate and Longitudinal pairs), we required: (i) *SNP Calling* filters be met, (ii) the number of reads aligning to the position is below the 99th percentile for all of the calls made for that isolate, (iii) the call at that position passes all filters for each isolate in the pair, and (iv) SNPs in *glpK* were dropped as mutants arising in this gene are thought to be an artifact of in vitro expansion (*Pethe et al., 2010*; *Vargas and Farhat, 2020*); we detected four non-synonymous SNPs in *glpK* ($\Delta AF \geq 25\%$) between three longitudinal pairs (mean $\Delta AF = 64\%$) and five non-synonymous SNPs in *glpK* ($\Delta AF \geq 25\%$) among five replicate pairs (mean $\Delta AF = 45\%$).

### Additional SNP filtering for antibiotic resistance loci analysis

To call SNPs (and corresponding minor changes in allele frequencies) between pairs of isolates (Longitudinal Pairs), we required: (i) *SNP Calling* filters be met, (ii) *Additional SNP Filtering for Isolate Pairs* filters be met, (iii) $\left| AF_1^{alt} - AF_2^{alt} \right| = \text{AF} \geq 5\%$, (iv) if $5\% \leq \text{AF} < 20\%$, then the SNP was only retained if each allele (across both isolates) with $\text{AF} > 0\%$ was supported by at least five reads (ensuring that at least five reads supported each minor allele at lower values of $\text{AF}$), (v) the SNP was classified as either intergenic or non-synonymous, (vi) the SNP was located in a gene, intergenic region or rRNA coding region associated with antibiotic resistance (*Supplementary file 5*).

### Additional SNP filtering for heterogenous SNPs

To call heterogenous SNPs in each isolate for a pair of longitudinal isolates, we required: (i) *SNP Calling* filters be met, (ii) the number of reads aligning to the position is below the 99th percentile for all of the calls made for that isolate, (iii) the alternate allele frequency for the SNP is such that $25\% \leq AF^{alt} \leq 75\%$, (iv) the position is not located in a mobile genetic element region of the reference genome, (v) the position is not located in a PE/PPE region of the reference genome.

### Additional SNP filtering for global isolates

To call alleles in our global and genetically diverse set of isolates, we required: (i) *SNP Calling* filters be met, with the modifications that (ii) the call was flagged as *Pass* by Pilon and (iii) 1 allele (either the reference or an alternate) was supported by at least 90% of the reads regardless of whether the other $\leq 10\%$ of reads supported 1, 2, or 3 alleles.

## Mixed lineage and contamination detection for longitudinal and replicate isolate pairs

### Kraken

To filter out samples that may have been contaminated by foreign DNA during sample preparation, we ran the trimmed reads for each longitudinal and replicate isolate through Kraken2 (*Wood and Salzberg, 2014*) against a database (*Goig et al., 2020*) containing all of the sequences of bacteria, archaea, virus, protozoa, plasmids, and fungi in RefSeq (release 90) and the human genome (GRCh38). We calculated the proportion reads that were taxonomically classified under the *Mycobacterium tuberculosis* Complex (MTBC) for each isolate and implemented a threshold of 95%. An isolate pair was dropped if either isolate had less than 95% of reads aligning to MTBC.

### F2

To further reduce the effects of contamination, we aimed to identify samples that may have been patient to inter-lineage mixture samples resulting from of a co-infection (F2). We computed the F2 lineage-mixture metric for each longitudinal and replicate isolate (*Figure 1B*). We wrote a custom script to carry out the same protocol for computing F2 as previously described (*Wyllie et al., 2018*). Briefly, the method involves calculating the minor allele frequencies at lineage-defining SNPs (*Coll et al., 2014*). From 64 sets of SNPs that define the deep branches of the MTBC (*Coll et al., 2014*), we considered the 57 sets that contain more than 20 SNPs to obtain better estimates of minor variation (*Coll et al., 2014*; *Wyllie et al., 2018*). For each SNP set $i$, (i) we summed the total depth and (ii) the number of reads supporting the most abundant base (at each position) over all of the reference positions (SNPs) that met our mapping quality, base quality and insertion/deletion filters, which yields $d_i$ and $x_i$ respectively. Subtracting these two quantities yields the minor depth for SNP set $i$, $m_i = d_i - x_i$. The minor allele frequency estimate for SNP set $i$ is then defined as $p_i = m_i d_i$. Doing this for all 57 SNP sets gives $\{p_1, p_2, \cdots, p_{57}\}$. We then sorted $\{p_1, p_2, \cdots, p_{57}\}$ in descending order and estimated the minor variant frequency for all of the reference positions (SNPs) corresponding to the top 2 sets (highest $p_i$ values) which yields the F2 metric. Letting $n2$ be the number of SNPs in the top two sets, then $F2 = \sum_{j=1}^{n2} m_j \ / \ \sum_{i=1}^{n2} d_i$. Isolate pairs were dropped if the F2 metric for either isolate passed the F2 threshold set for mixed lineage detection (*Figure 1B* and *Figure 1—figure supplement 1*).

## Pre-existing genotypic resistance

We determined pre-existing resistance for a patient (with a pair of longitudinal isolates) by scanning the first isolate for the detection of at least 1 of 177 SNPs predictive of resistance with AF $\geq$75% (from a minimal set of 238 variants [*Farhat et al., 2016*]). Drug resistance was inferred from the whole genome sequencing data using a well validated set of 177 mutations at an allele frequency threshold (>40%). Selection of these mutations and validation of this allele frequency threshold was previously described. This study made use of 1319 clinical Mtbc isolates with known drug resistance phenotypes. The data were randomly split into training and validation sets containing 67% and 33% of the isolates (respectively). The diagnostic set of mutations was determined using random forest predictive modeling in which a weighted model was run with serially smaller subsets of mutations to identify a minimal set of mutations to predict resistance to first- and second-line TB drugs. The resulting set of mutations predicted INH resistance with a sensitivity of 94% and specificity of 94% on the validation isolate set and predicted RIF resistance with a sensitivity of 93% and specificity of 95% on the validation isolate set. We excluded predictive indels and the *gid* E92D variant as the latter is likely a lineage marking variant that is not indicative of antibiotic resistance. We defined pre-existing multidrug resistance for a patient by scanning the first isolate collected for detection of at least 1 SNP predictive of Rifampicin resistance (14/178 predictive SNPs) and at least 1 SNP predictive of Isoniazid resistance (18/178 predictive SNPs). The genotypic resistance predictions for 13 antibiotics for all 614 longitudinal isolates from the 307 patients in our study can be found in *Supplementary file 21*.

## True and false positive rate analysis for heteroresistant mutations

To determine the predictive value of low-frequency heteroresistant alleles, we classified SNPs as fixed if the alternate allele frequency in the second isolate collected from the patient was at least

75% (alt $AF_2 \geq 75\%$). We first dropped SNPs for which alt $AF_1 \geq 75\%$ and alt $AF_2 \geq 75\%$ (high frequency mutant alleles in both isolates). We then set a threshold ($F_i$) for the alternate allele frequency detected in the first isolate collected from the patient (alt $AF_1$) and predicted whether an alternate allele would rise to a substantial proportion of the sample (alt $AF_2 \geq 75\%$) as follows:

$$alt\ AF_1 < F_i \longrightarrow alt\ AF_2 < 75\%$$

$$alt\ AF_1 \geq F_i \longrightarrow alt\ AF_2 \geq 75\%$$

We classified every SNP as True Positive (TP), False Positive (FP), True Negative (TN) or False Negative (FN) according to:

$$TP:\ alt\ AF_1 \geq F_i\ \ \&\ \ alt\ AF_2 \geq 75\%$$

$$FP:\ alt\ AF_1 \geq F_i\ \ \&\ \ alt\ AF_2 < 75\%$$

$$TN:\ alt\ AF_1 < F_i\ \ \&\ \ alt\ AF_2 < 75\%$$

$$FN:\ alt\ AF_1 < F_i\ \ \&\ \ alt\ AF_2 \geq 75\%$$

True Positive Rates (TPR) and False Positive Rates (FPR) were calculated as:

$$TPR = \frac{\#TP}{\#TP + \#FN} \quad FPR = \frac{\#FP}{\#FP + \#TN}$$

Finally, we made predictions for all SNPs and calculated the TPR and FPR for all values of $F_i \in \{0\%, 1\%,\ 2\%, \cdots, 98\%, 99\%, 100\%\}$.

## Mutation density test

The method to detect significant variation for a given locus amongst pairs of sequenced isolates has been described previously (*Farhat et al., 2014*). Briefly, let $\mathcal{N}_j \sim Pois(\lambda_j)$ be a random variable for the number of SNPs detected across all isolate pairs (for the in-host analysis this is the collection of longitudinal isolate pairs for all patients) for gene $j$. Let (i) $N_i$ = number of SNPs across all pairs for gene $i$, (ii) $|g_i|$ = length of gene $i$, (iii) $P$ = number of genome pairs and (iv) $G$ = the number of genes across the genome being analyzed (all genes in the essential, non-essential, antigen, antibiotic resistant and family protein categories).

Then the length of the genome (concatenate of all genes being analyzed) is given by $\sum_{i=1}^{G} |g_i|$ and the number of SNPs across all genes and genome pairs is given by $\sum_{i=1}^{G} N_i$. The null rate for $\mathcal{N}_j$ is given by the mean SNP distance between all pairs of isolates, weighted by the length of gene $j$ as a fraction of the genome concatenate and number of isolate pairs:

$$\lambda_j = \left( \frac{\sum_{i=1}^{G} N_i}{P} \right) \left( \frac{|g_i|}{\sum_{i=1}^{G} |g_i|} \right) \left( \frac{1}{P} \right)$$

The p-value for gene $j$ is then calculated as $\Pr(N_i > \mathcal{N}_j)$. We tested 3,386 genes for mutational density and applied Bonferroni correction to determine a significance threshold. We determine a gene to have a significant amount of variation if the assigned p-value $< \frac{0.05}{3,386} \approx 1.477 \times 10^{-5}$.

## Nucleotide diversity

We define the nucleotide diversity ($\pi_g$) for a given gene $g$ as follows: (i) let $|gene_g|$ = base-pair length of the gene, (ii) $N_{i,j}$ = number of in-host SNPs (independent of the change in allele frequency for each SNP) between the longitudinal isolates for patient $i$ occurring on gene $j$ and (iii) $P$ = number of patients. Then

$$\pi_g = \left( \frac{1}{P} \right) \left( \frac{1}{|gene_g|} \right) \sum_{i=1}^{P} N_{i,g}$$

Correspondingly, let $G$ be a category consisting of $M$ genes, then the average nucleotide diversity for $G$ is given by:

$$\pi_G = \left(\frac{1}{M}\right)\left(\frac{1}{P}\right)\sum_{j=1}^{M}\left(\frac{1}{|gene_j|}\right)\left(\sum_{i=1}^{P}N_{i,j}\right)$$

## SNP calling simulations in repetitive genomic regions

Certain repetitive regions of the *Mycobacterium tuberculosis* genome (ESX, PE/PPE loci) may give rise to false positive and false negative variant calls due to the mis-alignment of short-read sequencing data. To test the rate of false negative and false positive SNP calls in genes with *in-host* SNPs (*Figure 5*, *Supplementary file 8*) we collected the set of non-redundant SNPs observed in these loci (*Supplementary file 10*). Next, we collected a set of publicly available reference genomes (*Supplementary file 9*) and introduced these mutations into the respective loci positions in the reference genomes. We then simulated short-read Illumina sequencing data of comparable quality to our sequencing data from these altered reference genomes. Using our variant-calling pipeline to call polymorphisms, we then estimated the number of true and false positive SNP calls for each gene, based off of how many introduced SNPs were called (true positives), how many introduced SNPs were not called (false negatives) and how many spurious SNPs were called (false positives). A schematic of our simulation methodology is given in *Appendix 1—figure 1, a* detailed explanation is given in Appendix 1 and the results of our simulations (given in *Appendix 1—figure 2*) confirm a low false-positive rate.

## Global lineage typing

We determined the global lineage of each longitudinal ($N = 614$) and global isolate ($N = 32,033$) using base calls from Pilon-generated VCF files and a 62-SNP lineage-defining diagnostic barcode from a previously published study (*Coll et al., 2014*).

## Phylogenetic convergence analysis and t-SNE visualization

### Construction of genotypes matrix

We detected SNP sites at 878,244 H37Rv reference positions (of which 61,918 SNPs were not biallelic) among our global sample of 33,873 isolates. After excluding SNP sites with rare minor alleles (sites in which alternate alleles were called in <5 isolates) we retained SNPs at 146,874 positions. We constructed a 146,874 × 33,873 genotypes matrix (coded as 0:A, 1:C, 2:G, 3:T, 9:Missing) and filled in the matrix for the allele supported at each SNP site for each isolate according to the *SNP Calling* filters outlined above. If a base call at a specific reference position for an isolate did not meet the filter criteria that allele was coded as *Missing*.

Excluding 2348 SNP sites that had an EBR score <0.80, another 1509 SNP sites located within mobile genetic element regions, and 3220 SNP sites in with missing calls in >25% of isolates yielded a genotypes matrix with dimensions 139,797 × 33,873. Next, we excluded 1518 isolates with missing calls in >25% of SNP sites yielding a genotypes matrix with dimensions 139,797 × 32,355. We used a previously published 62-SNP barcode (*Coll et al., 2014*) to type the global lineage of each isolate in our sample. We further excluded 322 isolates that did not get assigned a global lineage, another 100 isolates that were used in our longitudinal analysis (*Supplementary file 3*), 152 isolates typed as *Mycobacterium bovis*, and 35 isolates typed as lineage 7. To improve computational efficiency and runtime, we randomly down sampled (using Python's random.sample() function) our collection of lineage 2 and lineage 4 isolates to include 7000 isolates of each lineage in our sample. This excluded 1064 lineage 2 isolates and 10,330 lineage four isolates. These steps yielded a genotypes matrix with dimensions 139,797 × 20,352. Finally, we excluded 10,899 SNP sites from this filtered genotypes matrix in which the minor allele count = 0. The genotypes matrix used for downstream analysis had dimensions 128,898 × 20,352 representing 128,898 SNP sites across 20,352 isolates. The global lineage breakdown of the 20,352 isolates was: L1 = 2802, L2 = 7000, L3 = 3352, L4 = 7,000, L5 = 101, L6 = 97.

## Phylogenetic convergence test

We tested 141/174 in-host SNPs in which the alternate (mutant) allele frequency increased substantially across sampling (*Supplementary file 8*) for phylogenetic convergence. We scanned a set 20,352 global and genetically diverse isolates from our 128,898 × 20,352 genotypes matrix for these SNPs (*Supplementary file 18*). To determine phylogenetic convergence for a given SNP site, we required that the alternate allele (called against the H37Rv reference) be present but not fixed in at least three global lineages. More specifically, we required that the alternate allele be detected in at least one isolate within each lineage and not in >95% of the isolates in that lineage. Twenty-six SNP sites across 12 genes and three intergenic regions were detected as having a signal of phylogenetic convergence (*Supplementary file 19*).

## t-SNE visualization

To construct the t-SNE plots that captured the genetic relatedness of the 20,352 isolates in our sample, we first constructed a pairwise SNP distance matrix. To efficiently compute this using our 128,898 × 20,352 genotypes matrix, we binarized the genotypes matrix and used sparse matrix multiplication implemented in Scipy to compute five 20,352 × 20,352 similarity matrices (*Virtanen et al., 2020*). We constructed a similarity matrix for each nucleotide (*A, C, G, T*) where row *i*, column *j* of the similarity matrix for nucleotide *x* stored the number of *x*'s that isolate *i* and isolate *j* shared in common across all SNP sites. The fifth similarity matrix (*N*) stored the number of SNP sites in which neither isolate *i* and isolate *j* had a missing value. The pairwise SNP distance matrix (*D*) was then computed as $D = N - (A + C + G + T)$. $D$ had dimensions 20,352 × 20,352 where row *i*, column *j* stored the number of SNP sites in which isolate *i* and isolate *j* disagreed. We used $D$ as input into a t-SNE algorithm implemented in Scikit-learn (*Pedregosa et al., 2011*) (settings: perplexity = 175, n_components = 2, metric = 'precomputed', n_iter = 1000, learning_rate = 1500) to compute the embeddings for all 20,352 isolates in our sample. We used these embeddings to visualize the genetic relatedness of the isolates in two dimensions and colored isolates (points on the t-SNE plot) by lineage (*Figure 7A*). For visualizing specific mutations, isolates were colored according to whether or not the alternate (mutant) allele was called (*Figure 7B–F*, *Figure 7—figure supplements 1–2*).

## Data analysis and variant annotation

Data analysis was performed using custom scripts run in Python and interfaced with iPython (*Perez and Granger, 2007*). Statistical tests were run with Statsmodels (*Seabold and Perktold, 2010*) and Figures were plotted using Matplotlib (*Hunter, 2007*). Numpy (*van der Walt et al., 2011*), Biopython (*Cock et al., 2009*) and Pandas (*McKinney, 2010*) were all used extensively in data cleaning and manipulation. Functional annotation of SNPs was done in Biopython (*Cock et al., 2009*) using the H37Rv reference genome and the corresponding genome annotation. For every SNP called, we used the H37Rv reference position provided by Pilon (*Walker et al., 2014*) generated VCF file to extract any overlapping CDS region and annotated SNPs accordingly. Each overlapping CDS regions was then translated into its corresponding peptide sequence with both the reference and alternate allele. SNPs in which the peptide sequences did not differ between alleles were labeled *synonymous*, SNPs in which the peptide sequences did differ were labeled *non-synonymous* and if there were no overlapping CDS regions for that reference position, then the SNP was labeled *intergenic*.

## Pathway definitions

We used SEED (*Overbeek et al., 2014*) subsystem annotation to conduct pathway analysis and downloaded the subsystem classification for all features of *Mycobacterium tuberculosis* H37Rv (id: 83332.1) (*Supplementary file 15*). We mapped all of the annotated features from SEED to the annotation for H37Rv. Due to the slight inconsistency between the start and end chromosomal coordinates for features from SEED and our H37Rv annotation, we assigned a locus from H37Rv to a subsystem if both the start and end coordinates for this locus fell within a 20 base-pair window of the start and end coordinates for a feature in the SEED annotation (*Supplementary file 16*).

## Data and materials availability

All Mtbc sequencing data was collected from previously published studies and is publicly available. Individual accession numbers for the Mtbc genomes analyzed in this study can be found in *Supplementary file 2* and information on which studies from which the data was generated can be found in the Materials and methods, *Figure 1—figure supplement 1* and *Supplementary file 1*. All packages and software used in this study have been noted in the Materials and methods. Custom scripts written in python version 2.7.15 were used to conduct all analyses and interfaced via Jupyter Notebooks. Jupyter Notebooks and scripts written for data processing and analysis can be found in the following GitHub repository - https://github.com/farhat-lab/in-host-Mtbc-dynamics; *Vargas, 2021* (copy archived at swh:1:rev:36e27011b5cfaed00521a38652fe2dc853832f25).

## Acknowledgements

We thank the members of the Farhat lab for helpful discussions and comments on the research project and manuscript. We thank S Fortune, N Hicks and D Warner for helpful suggestions on the manuscript. We thank A Narayan for helpful suggestions on constructing t-SNE visualizations for phylogenetic convergence. RVJ was supported by the National Science Foundation Graduate Research Fellowship under Grant No. DGE1745303. MF was supported by NIH/BD2K K01 ES026835 and NIH NIAID R01 AI55765. The content is solely the responsibility of the authors and does not necessarily represent the official views of the National Institutes of Health. Portions of this research were conducted on the O2 High Performance Compute Cluster, supported by the Research Computing Group, at Harvard Medical School.

## Additional information

### Funding

| Funder | Grant reference number | Author |
|---|---|---|
| National Science Foundation | DGE1745303 | Roger Vargas |
| National Institutes of Health | K01 ES026835 | Maha Reda Farhat |
| National Institutes of Health | R01 AI55765 | Maha Reda Farhat |

The funders had no role in study design, data collection and interpretation, or the decision to submit the work for publication.

### Author contributions

Roger Vargas, Conceptualization, Resources, Data curation, Software, Formal analysis, Validation, Investigation, Visualization, Methodology, Writing - original draft, Writing - review and editing; Luca Freschi, Maximillian Marin, Data curation, Bioinformatics Support; L Elaine Epperson, David Durbin, Michael Strong, Max Salfinger, Resources, Cultured Mtb isolates and performed DNA extraction in preparation for PacBio sequencing; Melissa Smith, Irina Oussenko, Resources, Prepared libraries and performed PacBio sequencing runs; Maha Reda Farhat, Conceptualization, Resources, Data curation, Formal analysis, Supervision, Investigation, Visualization, Methodology, Writing - original draft, Writing - review and editing

### Author ORCIDs

Roger Vargas (iD) https://orcid.org/0000-0002-7116-5211

### Decision letter and Author response

Decision letter https://doi.org/10.7554/eLife.61805.sa1
Author response https://doi.org/10.7554/eLife.61805.sa2

# Additional files

## Supplementary files

• Supplementary file 1. A table containing details for the eight studies; the sources for the longitudinal isolate pairs. Information includes: (1) reference for each source study, (2) number of patients included in this study, (3) a description of the sample collection, (4) timing of when sputum samples were collected relative to treatment initiation/cessation (if available).

• Supplementary file 2. A table containing details for all replicate and longitudinal isolates before Kraken, F2, or pairwise SNP filtering. Includes aggregated patient treatment from the source studies and metadata for each longitudinal isolate. We include columns that indicate the timing of sampling of Mtbc relative to treatment, the treatment regimen administered and final patient outcome (and relevant details). Patient outcomes are defined as follows: *Delayed culture conversion* (sputum culture positive at baseline and 2 months treatment initiation with genomic analysis consistent with clonal infection), *Failure or Relapse* (sputum culture positive at baseline and 4.5 months treatment initiation with genomic analysis consistent with clonal infection), *Failure or Relapse or Default* (sputum culture positive at interval of 4.5 months with genomic analysis consistent with clonal infection, only partial treatment data is available) or N/A if date data was of low resolution, not available or no treatment data was available. We also determined *Reinfection* and *Mixed infection* based on the genomic analysis. Legend for antibiotics: H = isoniazid, R = rifampicin, Rp = rifapentine, E = ethambutol, Z = pyrazinamide, C = capreomycin, S = cycloserine, P = para-aminosalicylic acid, Et = ethionamide, K = kanamycin, A = Amikacin, L = levofloxacin, Cf = ciprofloxacin, M = moxifloxacin, Sm = streptomycin.

• Supplementary file 3. A table containing details for all $(n = 400)$ longitudinal isolates used for in-host analysis after filtering for contaminated and mixed isolate pairs.

• Supplementary file 4. A table with the gene categories assigned to each H37Rv locus tag.

• Supplementary file 5. A table containing a list of genomic regions (with H37Rv coordinates) associated with antibiotic resistance.

• Supplementary file 6. A table containing all SNPs (with $\Delta AF \geq 5\%$) in loci associated with antibiotic resistance (*Supplementary file 5*) across our sample of 200 longitudinal isolate pairs.

• Supplementary file 7. A table containing all pre-existing antibiotic resistant SNPs detected in the first isolate collected from each patient with collection dates ≥2 months apart (178/200).

• Supplementary file 8. A table containing information for all 174 in-host SNPs detected across all longitudinal isolate pairs.

• Supplementary file 9. A table with details for the 54 publicly available completed (reference) genomes used in our simulations.

• Supplementary file 10. A table with the non-redundant *in-host* SNPs identified within genes and used for SNP calling simulations.

• Supplementary file 11. A table containing all of the epitopes downloaded from IEDB on May 23, 2018.

• Supplementary file 12. A table containing the epitopes belonging to *PPE18* where an in-host SNP was detected.

• Supplementary file 13. A table of all genes identified as *dense*, along with assigned gene category and p-value from mutation density test.

• Supplementary file 14. A table of all genes identified as *convergent*, along with assigned gene category and the number of patients with an in-host SNP in each gene.

• Supplementary file 15. A table containing the downloaded SEED annotation for H37Rv.

• Supplementary file 16. A table containing the list of H37Rv locus tags corresponding to each subsystem classified by SEED.

• Supplementary file 17. A table containing the pathways and (corresponding in-host SNPs) displaying evidence of parallel evolution.

- Supplementary file 18. A table with details for all SNP calls made in a global collection of 20,352 publicly available isolates after screening for in-host SNPs (*Supplementary file 8*).

- Supplementary file 19. A table with details for in-host SNPs (*Supplementary file 8*) that displayed a signature of phylogenetic convergence after screening a global collection of 20,352 publicly available isolates (Materials and methods). The number of isolates with each unique mutation (broken down by global lineage) is given.

- Supplementary file 20. A table containing details for isolates that underwent Illumina and PacBio sequencing.

- Supplementary file 21. A table containing the genotypic resistance predictions for 13 antibiotics for all 614 longitudinal isolates from the 307 patients in our study (S: susceptible, R: resistant). Legend for antibiotics: INH = isoniazid, RIF = rifampicin, EMB = ethambutol, PZA = pyrazinamide, CAP = capreomycin, PAS = para-aminosalicylic acid, ETH = ethionamide, KAN = kanamycin, AMK = Amikacin, LEVO = levofloxacin, CIP = ciprofloxacin, OFLX = ofloxacin, STR = streptomycin.

- Transparent reporting form

### Data availability

All Mtbc sequencing data was collected from previously published studies and is publicly available. Individual accession numbers for the Mtbc genomes analyzed in this study can be found in Supplementary File 2 and information on which studies from which the data was generated can be found in the Methods, Figure 1 - figure supplement 1 and Supplementary File 1. All packages and software used in this study have been noted in the Methods. Custom scripts written in python version 2.7.15 were used to conduct all analyses and interfaced via Jupyter Notebooks. Jupyter Notebooks and scripts written for data processing and analysis can be found in the following GitHub repository - https://github.com/farhat-lab/in-host-Mtbc-dynamics (copy archived at https://archive.softwareheritage.org/swh:1:rev:36e27011b5cfaed00521a38652fe2dc853832f25/).

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

# Appendix 1

## SNP calling simulations

### Reference genome collection

We downloaded 60 reference genomes (RefGenome) (i.e. completely assembled *Mycobacterium tuberculosis* genomes) from NCBI (RRID:SCR_002760) (Genbank accession IDs can be found in *Supplementary file 9*). We limited our collection to genomes for which there were corresponding annotation files.

### Mapping CDS regions from reference genomes to H37Rv

Since the regions of interest were repetitive loci that have many homologies elsewhere in the genome, we were unable to use traditional alignment methods to map the genes of interest from H37Rv to the other RefGenomes. Instead, we made use of the clonal structure of the Mtbc genome to construct gene mappings from H37Rv to the RefGenomes as follows (*Appendix 1—figure 1A*):

1. For each gene $g$ annotated in H37Rv, collect the set of gene lengths 5 genes upstream and 5 genes downstream of $g$ from H37Rv. Compare the set of 11 H37Rv gene lengths to every set of 11 consecutive gene neighborhoods on the RefGenome and assign a score based off of the intersection of each pair of sets.
2. Look at the gene neighborhood(s) with the top score after scanning the RefGenome and pairwise globally align (*Cock et al., 2009*) $g$ to every gene in the top scoring neighborhood using the following criteria: (i) identical characters are given 2 points, (ii) 1 point is deducted for each non-identical character, (iii) 2 points are deducted for opening a gap, (iv) 2 points are deducted for extending a gap.
3. Take the top scoring alignment $r$ and assign a mapping from H37Rv gene $g$ to RefGenome gene $r$ if (i) the pairwise alignment score is >0 and (ii) the base pair length of $g$ and $r$ are equivalent (the latter ensures correct placement of mutations in downstream analysis). If either of these criteria is not met, then we do not assign a mapping from $g$ to any CDS region on that RefGenome.

### Filtering low-quality mapped reference genomes

To assess the quality of the mappings from H37Rv to the set of RefGenomes, we compared the reference position start coordinates of each assigned mapping between each RefGenome and H37Rv. Again, making use of Mtbc clonality, we reasoned that the genomic structure of each pair of genomes is similar (if each RefGenome is indexed to start at the first gene on H37Rv *Rv0001*, then well mapped RefGenomes will have mapped genes that are located within a neighborhood of the coordinates from H37Rv). To test this (for each RefGenome), we took the absolute difference between the start coordinates for all of the mapped genes between the RefGenome and H37Rv. We then averaged these differences across all gene mappings between both genomes.

This measures the conservation (of the ordering) of the mapped genes between each pair of genomes (H37Rv and RefGenome) and gives an indication of how successful the mappings were on a global scale. We downloaded and mapped genes for 60 Genome Assemblies from GenBank (*Benson et al., 2009*) and assessed the quality of each set of mappings using the measure described above (*Appendix 1—figure 1B-C*). We excluded 6 RefGenomes on the basis of sporadic gene mappings against H37Rv which was determined by looking at the distribution of the mapping measure for all 60 assemblies. We kept the remaining 54 genomes for use in the simulations (*Supplementary file 9*).

### Altering RefGenomes at SNP test sites

We make use of the set of the (non-redundant) observed in-host SNPs across all genes (*Figure 5D*, *Supplementary file 10*). We alter each RefGenome by introducing mutations (that correspond to the aforementioned SNPs) into the genes successfully mapped to H37Rv, ensuring that the new bases differ from the corresponding base positions on H37Rv. Since successful mappings require that the mapped genes be the same length, the mutations are introduced into the same site on the

RefGenome with respect to the gene specific coordinates (i.e. a gene $n$ bp long will have coordinates $\{1, 2, \cdots, n-1, n\}$ from $5' \to 3'$). We store information pertaining to which bases were altered for each RefGenome $\{SNP\ set\ \beta\}$. No simulations are run for genes on RefGenomes that are not successfully mapped to H37Rv.

## Simulating reads from complete Genomes

To validate our SNP calling methodology using the set of RefGenomes, we used ART (*Huang et al., 2012*) to simulate short-read sequencing data altered versions of the RefGenomes (*Appendix 1—figure 1B*). Since the aim of our simulations was to study the quality of our variant calls on our real data, we simulated data for each (altered) RefGenome that was of comparable quality to our real sequencing data: Illumina HiSeq 1000, read length of 100 bp, mean coverage of 80x, paired end reads, 200 bp mean size of DNA fragments, 25 bp standard deviation of DNA fragment size (settings: -ss HS10 -l 100 f 80 p -m 200 s 25).

## Mapping simulated reads to H37Rv and calling SNPs

Next we mapped the pool of simulated reads from the altered RefGenomes against the H37Rv reference genome and called SNPs according to most of the same procedures and WGS filters outlined in Materials and methods. However, in this instance we called SNPs at reference positions that supported an alternate allele and required that calls were flagged as *Pass* by Pilon (where the alternate allele frequency was $\geq$75% and no *Ambiguous*, *Low Coverage*, or *Deletion* flags were present at that position). For each RefGenome, this yielded the set of SNPs (between the altered RefGenome and H37Rv) called by our pipeline $\{SNP\ set\ \mathbf{B}\}$ (*Appendix 1—figure 1B*).

## Calling SNPs with MUMmer

We used Mummer3 (*Kurtz et al., 2004*) to call SNPs between H37Rv and each (unaltered) RefGenome. We aligned each pair of genomes and called SNPs between the alignments using the following commands:

1. `nucmer -mum H37Rv.fasta RefGenome.fasta`
2. `delta-filter -r -q H37Rv_RefGenome.delta > H37Rv_RefGenome.filter`
3. `show-snps -Clr -T H37Rv_RefGenome.filter > H37Rv_RefGenome.snps`

The resulting SNP calls yielded the set of SNPs between each of the unmodified (unaltered) RefGenomes and H37Rv $\{SNP\ set\ \mathbf{A}\}$ (*Appendix 1—figure 1B*).

## True and false positive SNP call analysis

To calculate the number of *true positives* and *false positives* with regard to our SNP calling pipeline for each gene $g$ of interest (*Appendix 1—figure 2*), we define the following sets of H37Rv coordinates for each RefGenome:

- $\beta$ - SNPs introduced into (altered) RefGenome
- $A$ - SNPs called between (unaltered) RefGenome & H37Rv
- $B$ - SNPs called between (altered) RefGenome & H37Rv
- $C$ - all reference positions (or coordinates) on H37Rv

The set of coordinates where an alternate allele was introduced into the RefGenome and called by the pipeline (true positive SNPs for gene $g$) is given by:

$$TP_g = \left(B_g A_g\right) \cap \left(\beta_g A_g\right)$$

where we normalize by SNP set $A_g$ to make sure we're only accounting for test SNPs in our computations. The set of coordinates where an alternate allele was note introduced and called by the pipeline (false positive SNPs for gene $g$) is given by:

$$FP_g = \left(\left(B_g A_g\right) \cap C_g\right) TP_g$$

The set of coordinates where an alternate allele was introduced but was not called by the pipeline (false negative SNPs for gene $g$) is given by:

$$FN_g = (\beta_g A_g) TP_g$$

The results of our simulations (*Appendix 1—figure 2*) indicate that the number of true positive calls is consistent with the number of known SNPs across all genes and simulations. Perhaps more importantly, our results also suggest that false positive calls are rarely made for any SNP in our sample. Thus, while we may not have called all of the existing variation between paired isolates (false negative calls), it is unlikely that we called non-existing variation between any pair of isolates (false positives). That is, false-positive SNPs are rarely called, even in repetitive loci such as the PE/PPE gene family, supporting our decision to keep all SNP calls for downstream analysis.

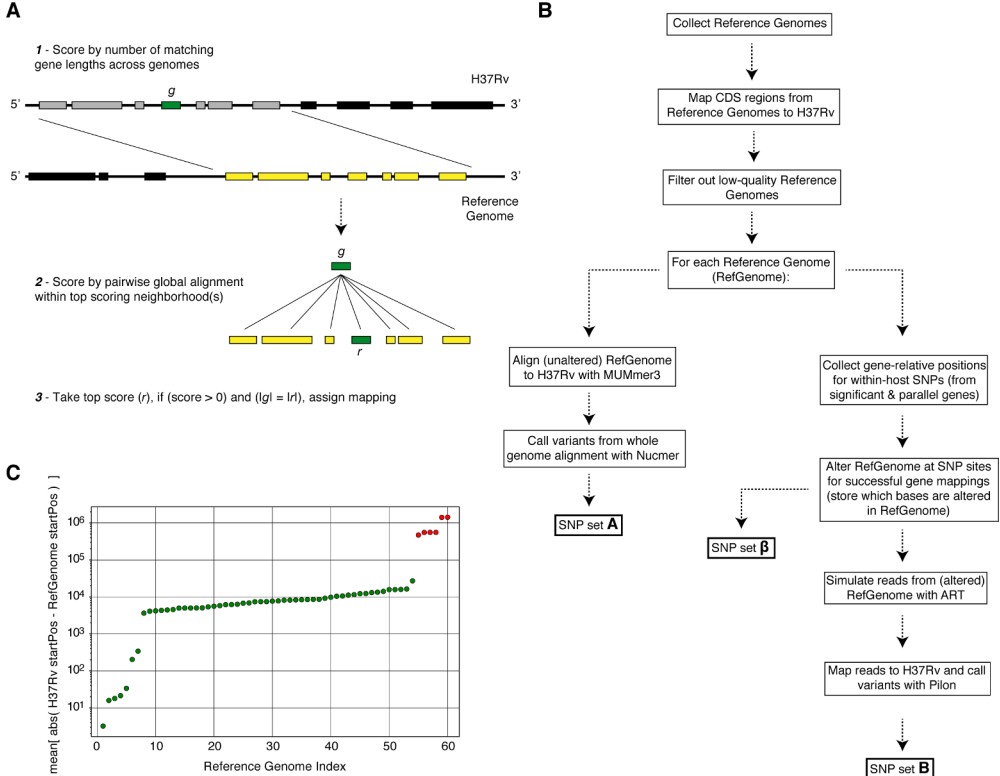

**Appendix 1—figure 1.** Overview of simulation methodology. To test the accuracy of calling SNPs in repetitive regions with our workflow, we introduced mutations into complete *Mycobacterium tuberculosis* genomes (Reference Genomes), simulated reads from those genomes and assessed the accuracy recalling the mutations from the simulated reads while not introducing spurious mutations. (**A**) We used a sliding window of gene lengths along with a local alignment algorithm to map genes from the H37Rv reference genome to the set Reference Genomes. (**C**) We discarded Reference Genomes that mapped poorly (gene-to-gene) to the H37Rv reference genome (green-RefGenomes kept for simulations, red-discarded RefGenomes). (**B**) A schematic of our simulation methodology from Reference Genome collection to obtaining SNP sets $A$, $B$ and $\beta$ which are used in our calculations of true positive and false positive calls for each gene.

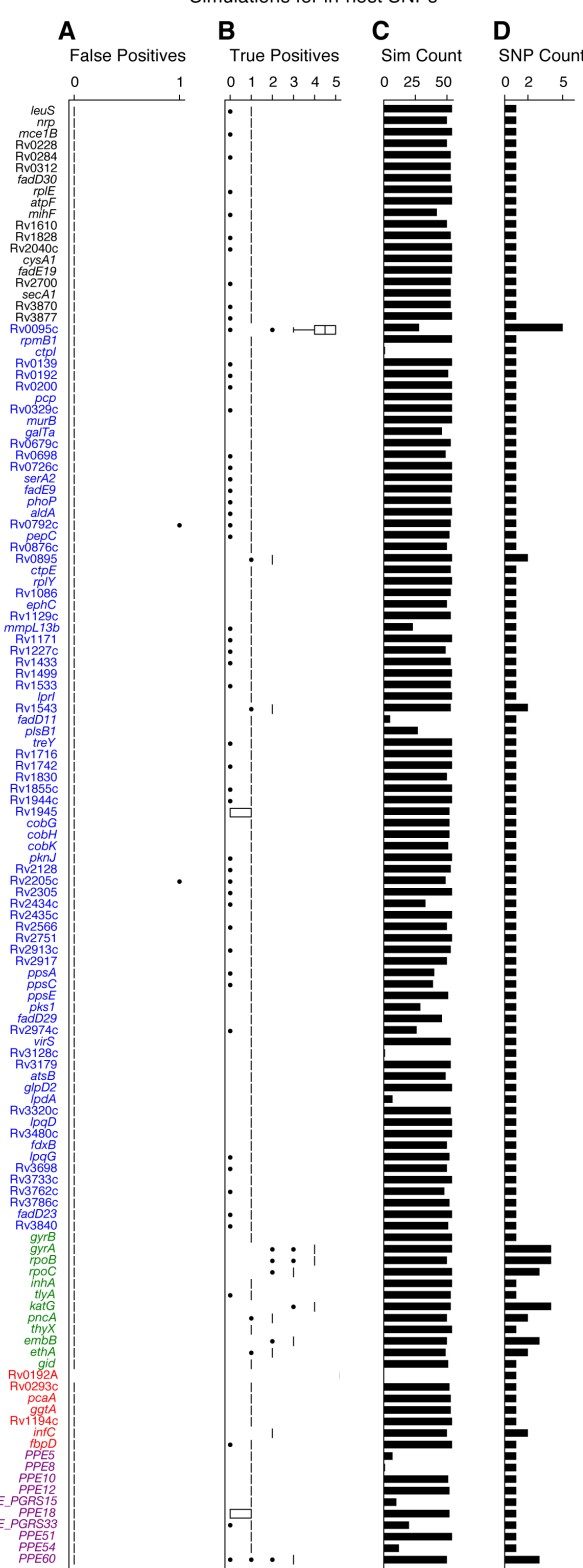

**Appendix 1—figure 2.** Simulations indicate that we can accurately recall most introduced SNPs while rarely making spurious SNP calls. We tested the number of true and false positives for each gene with detectable in-host SNPs (*Figure 5D*). For each gene we collected a set of non-redundant

*Appendix 1—figure 2 continued on next page*

*Appendix 1—figure 2 continued*

*in-host* SNPs (genomic positions at which these SNPs were called) observed across all patients (*Supplementary file 10*), the number of SNPs collected for each gene is given in (**D**). We then introduced these mutations into 54 complete genomes (RefGenomes) (*Supplementary file 9*) and simulated reads after introducing the respective mutations. Only genes that were mapped from H37Rv to a given RefGenome were part of the simulation for that RefGenome. (**C**) The number of successful mappings for each gene (i.e. the number of times each gene was part of a simulation). This is also the number of times true and false positive estimates were calculated for each gene (one estimate / simulation). (**A**) False positive calls were rarely made across all genes and simulation runs indicating the rarity of false positive SNP calls (calling a mutation that wasn't introduced) made by our pipeline for observed in-host SNPs, even in repetitive regions. (**B**) The number of true positive calls across all genes (across most simulation runs) closely matched the number of introduced SNPs for each gene indicating the rarity of False Negative SNP calls (not calling a mutation that was introduced). We note that no true or false positive estimates for *Rv0192A* were computed since this gene did not map to H37Rv for any of the 54 Reference Genomes used for the simulations.

## Appendix 2

### PacBio assembly vs. Illumina mapping SNP calling

DNA extraction and PacBio sequencing of Mtbc isolates

DNA extraction was performed according to a published protocol (*Epperson and Strong, 2020*). Approximately 1 μg of high molecular weight genomic DNA was used as input for SMRTbell preparation, according to the manufacturer's specifications (SMRTbell Template Preparation Kit 1.0, Pacific Biosciences, https://www.pacb.com/wp-content/uploads/2015/09/Procedure-Checklist-20-kb-Template-Preparation-Using-BluePippin-Size-Selection.pdf). Briefly, HMW gDNA was sheared to 20 kb using the Covaris g-tube at 4500 rpm. Following shearing, gDNA underwent DNA damage repair, ligation to SMRTbell adaptors and exonuclease treatment to remove any unligated gDNA. At least 500 ng final SMRTbell library per sample was cleaned with AMPure PB beads and 3-50 kb fragments were size selected using the BluePippin system on 0.75% agarose cassettes and S1 ladder, as specified by the manufacturer (Sage Science). Size selected SMRTbell libraries were annealed to sequencing primer and bound to the P6 polymerase prior to loading on the RSII sequencing system (Pacific Biosciences). Sequencing was performed using C4 chemistry and 240-min movies. Following data collection, raw data was converted into subreads for subsequent analysis using the RS_Subreads.1 pipeline within SMRTPortal (version 2.3), the web-based bioinformatics suite for analysis of RSII data.

PacBio de novo assembly, genome polishing, and variant calling

PacBio and Illumina sequencing data was available for 34 clinical Mtbc isolates (*Chiner-Oms et al., 2019*). We used Flye (*Kolmogorov et al., 2019*) to de novo assemble the raw PacBio subreads from these 34 isolates (settings: –pacbio-raw –genome-size 5 m) (version 2.5). If Flye identified the presence of a circular contig, Circlator (*Hunt et al., 2015*) was used to set the start each assembly at the DnaA locus. PacBio's bax2bam function (settings: –subread) was used to convert PacBio legacy BAX files to BAM format. We ran PacBio's implementation of Minimap2 (*Li, 2018*) (pbmm2) to map and sort raw PacBio subreads to the de novo assembly. We iteratively polished the assembly three times by running the Quiver algorithm (*Chin et al., 2013*) and used Samtools (*Li et al., 2009*) to index the fasta files from the resulting assemblies. Thirty-one of our 34 samples assembled into a single circular contig (*Supplementary file 20*). We excluded three isolates that did not have a single circular assembly from downstream analysis. To call SNPs relative to the H37Rv reference, we used Minimap2 (*Li, 2018*) to align each PacBio assembly to the H37Rv reference sequence. We used the *paftools.js call* utility included with Minimap2 to generate variant calls from each assembly to reference alignment.

## Appendix 3

### Antibiotic resistance analyses for confirmed failure and relapse patients

We repeated our analyses on the allele frequency dynamics within antibiotic resistance (AR) loci and rates of resistance amplification using a subset of 121/200 patients with confirmed treatment failure or relapse (*Supplementary file 2*) (corresponding to the following sections - Results: In-host pathogen dynamics in antibiotic resistance loci, Allele frequency >19% predicts subsequent fixation of resistance variants, Determinants of antibiotic resistance acquisition and microbiological treatment failure). For all 200 cases the order of sampling was available, but for 195/200 (119/121 confirmed failure patients) we also had the exact dates of sampling which were required for some analyses. We found that the analysis conclusions were unchanged between both the 121 subset and the full 200 patient sample.

### In-host pathogen dynamics in antibiotic resistance loci

We detected 1401 (compared to 1939 using 200 patients, *Figure 2B*) non-synonymous and intergenic SNPs in 36 antibiotic resistance loci (*Supplementary file 5*) that change in allele frequency by at least 5% between the first and second time points across our sample of 121 patients (*Appendix 3—figure 1A*). Of these SNPs, 1292 were non-synonymous, 61 were intergenic and 48 occurred with *rrs* (compared to 1774 non-synonymous, 91 intergenic, and 74 from the sample of 200 patients). We then determined the lowest AR frequency that can accurately predict the development of fixed resistance alleles later in time. After discarding 14 (20 using 200 patients) SNPs that were fixed (allele frequency >75%) at both times points, we studied the allele frequency trajectories of 1387 (1,919 using 200 patients) AR SNPs.

### Allele frequency >19% predicts subsequent fixation of resistance variants

We calculated the allele frequency AF1 (in the first isolate collected from each patient) that predicted subsequent fixation of resistance variants and allowed a maximum False Positive Rate of 5%. Using the full set of 200 patients we calculated an optimal threshold of AF1*=19% with an associated sensitivity of 27.0% and a specificity of 95.8% (*Figure 2C*). Ten mutant alleles across 14 isolates from seven patients had a frequency between 19% and 75% at the first time point and rose to fixation at the second time point (mean change in allele frequencies = 41%). Using the subset of 121 patients, we calculated a threshold of AF1*=17% with an associated sensitivity of 41.7% and a specificity of 95.3% (*Appendix 3—figure 1B*). Five mutant alleles across 10 isolates from four patients had a frequency between 17% and 75% at the first time point and rose to fixation at the second time point (mean change in allele frequencies = 48%).

### Determinants of antibiotic resistance acquisition and microbiological treatment failure

Analyzing the 119 patients with clonal infection and known isolate sampling dates (195 using the full set of patients), we aimed to identify overall rates of resistance acquisition by focusing on AR SNPs with moderate to high changes in allele frequency >= 40%. Using the set of 195 patients with clonal infection and sampling date, we detected 38 AR SNPs. We found that AR acquisition was more likely as the time between sampling increased, with the OR of AR acquisition being 1.023 per 30 day increment (95% CI 1.002, 1.045, p=0.035 Logistic Regression). Using the set of 119 patients with clonal infection and sampling date, we detected 13 AR SNPs. While AR acquisition was associated with the time between sampling, the association was not significant for this smaller sample with the OR acquisition being 1.017 per 30 day increment (95% CI 0.98, 1.055, p=0.375 Logistic Regression).

We then examined the relationship between pre-existing resistance and new AR acquisition in the subset of 195 patients that had samples collected >= 2 months apart consistent with persistent or relapsed infection for that duration (n = 178) and separately analyzed 119 patients with confirmed failure or relapse. We defined pre-existing resistance as >= 1 fixed AR SNP in the first isolate collected. Using the set of 178 patients, we found 259 pre-existing AR SNPs with 41% (73/178) of

failure patients harboring resistance to any drug at first sampling (*Figure 3B*). The majority of this resistance was MDR (*Figure 3C*) (multidrug resistance to at least isoniazid and rifamycin) 64% (47/73) and new resistance acquisition occurred mostly in patients with pre-existing resistance 20/27 (74%) (OR = 5.28, p=2.2×10-4 Fisher's exact test) or pre-existing MDR (OR = 3.85, p=3.4×10-3 Fisher's exact test). Using the set of 119 confirmed failure patients, we found 136 pre-existing AR SNPs with 30% (36/119) of failure patients harboring resistance to any drug at first sampling (*Appendix 3—figure 2A*). The majority of this resistance was MDR (*Appendix 3—figure 2B*) 67% (24/36) and new resistance acquisition occurred mostly in patients with pre-existing resistance 7/11 (64%) (OR = 4.77, p=1.8×10-2 Fisher's exact test) or pre-existing MDR (OR = 3.9, p=0.04 Fisher's exact test).

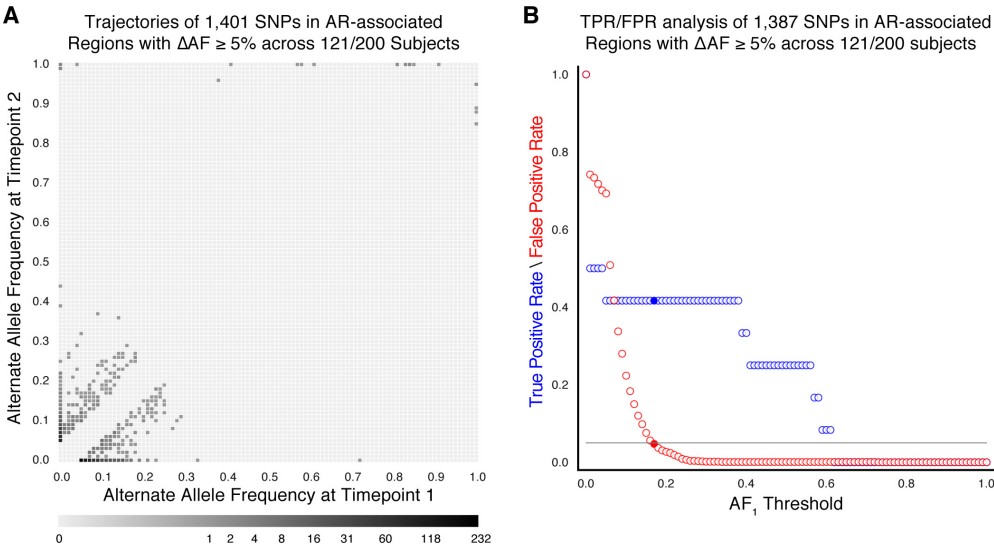

**Appendix 3—figure 1.** Allele frequency dynamics within antibiotic resistance loci in 121/200 confirmed failure/relapse patients. (**A**) The allele frequency trajectories for SNPs that occur in patients over the course of infection can be used to study the prediction of further antibiotic resistance using the frequency of alternate alleles detected in the longitudinal isolates collected from patients. (**B**) Plot of true positive rate (TPR) and false positive rate (FPR) for detecting eventual fixation of a resistance allele ($AF_2 \geq$ 75%) as a function of initial allele frequency ($AF_1$).

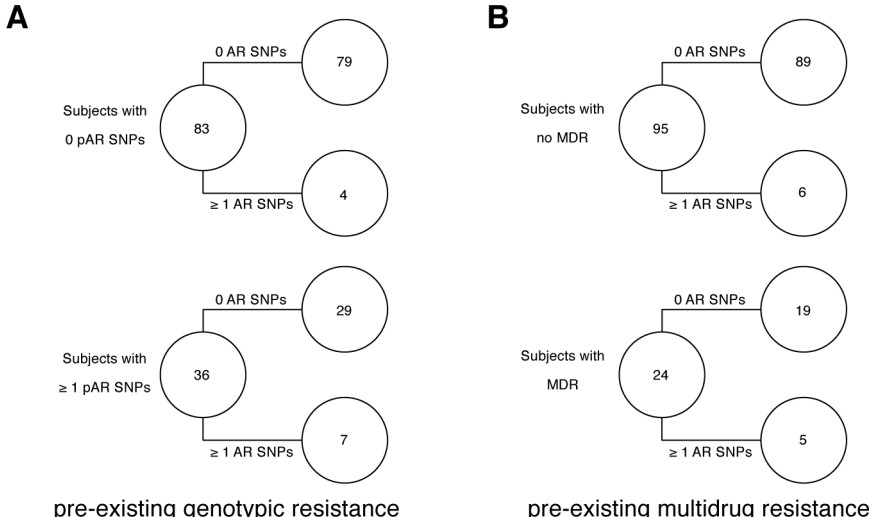

**Appendix 3—figure 2.** Pre-existing resistance is associated with resistance amplification in 121/200 confirmed failure/relapse patients. Among patients who fail treatment, (**A**) patients with pre-existing
*Appendix 3—figure 2 continued on next page*

*Appendix 3—figure 2 continued*

mutations that confer antibiotic resistance and (**B**) those that have pre-existing MDR are more likely to acquire antibiotic resistance mutations throughout the course of infection.

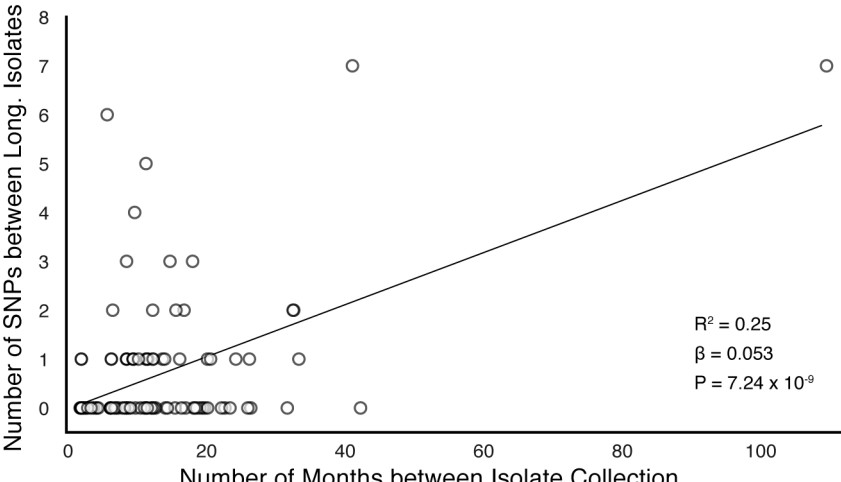

**Appendix 3—figure 3.** In-host SNP counts vs. time between isolate collection (119/121 confirmed failure/relapse patients with dates for both isolates shown). Regressing the number of SNPs per patients on the timing between isolate collection, we found SNPs to accumulate at an average rate of 0.64 SNPs per genome per year ($P = 7.24 \times 10^{-9}$).

