## [Decision Letter]

**Acceptance summary:**

Your manuscript provides interesting and novel insight on the level of within host genetic diversity displayed by *Mycobacterium tuberculosis* during colonisation of the human lung. Furthermore, the data on association of how the presence of these variants affect treatment outcome and the potential for emergence of drug resistance provide important insight, illustrating how pathogen genetics can be used to guide tuberculosis treatment.

**Decision letter after peer review:**

[Editors’ note: the authors submitted for reconsideration following the decision after peer review. What follows is the decision letter after the first round of review.]

Thank you for submitting your work entitled "In-host population dynamics of *M. tuberculosis* during treatment failure" for consideration by *eLife*. Your article has been reviewed by a Senior Editor, a Reviewing Editor, and three reviewers. The reviewers have opted to remain anonymous.

Our decision has been reached after consultation between the reviewers. Based on these discussions and the individual reviews below, we regret to inform you that your work will not be considered further for publication in *eLife*.

Reviewers considered the methodological approach in your study interesting and novel. However, there were several major concerns raised about select components of the analysis and appending conclusions. Further, the results did not appear to contribute a substantive advance over what is already known.

Reviewer #1:

The study aims to provide insights in the evolution of *M. tuberculosis* in the host with a particular focus on resistance development. The authors can show, that pre-existing mutations predict failure and can be fixed in follow up samples. That is per se not a new finding, but the data are comprehensive and provide a detailed view into in host dynamics in patients with failing regimens.

What I found problematic in the study is that the authors work with only two samples per patient, thus, not really allowing the analyses of the longitudinal evolutionary dynamics in TB patients. Accordingly, the title should be modified to reflect more the work the authors performed.

Also, the selection of the resistance variants during therapy has been shown several times in previous papers and is the natural process for resistance development. So, the information presented is not really new.

What I find interesting is the calculation of the "Lowest" low frequency level that can be detected. However, the paragraph is not really conclusive and the authors should refer to recently published papers e.g. Dreyer et al., 2020.

Overall, the paper is well written, however, more data esp. patient data need to be provided to allow a clear cut interpretation of the data presented.

Essential revisions

Introduction – I would refer to *M. tuberculosis* complex

Results

Subsection “Identifying clonal Mtb populations in-host” – where are the data concerning MTB lineage shown?

Subsection “In-host pathogen dynamics in antibiotic resistance loci” – how can you differentiate contaminations from real low frequency variants if you have only the two datapoints? What statistical method did you use to distinguish sequencing errors from real SNPs?

Subsection “Allele frequency >19% predicts subsequent fixation of resistance variants” – this paragraph is not really clear to me – what was the lowest frequency then that can be detected and results in treatment failure? Please look at Dreyer et al., 2020.

Subsection “Determinants of antibiotic resistance acquisition and microbiological treatment failure” – don't understand this sentence – If the patients have treatment failure, they are treated all the time – or?

Subsection “Determinants of antibiotic resistance acquisition and microbiological treatment failure” – the finding that pre-existing resistance mutations are leading to a higher chance of failure and resistance development is not surprising. What needs to be provided here are data on the effectivity of the given regimen. Meaning, one needs to know with how many active drugs a patient was treated when the low level resistance mutation was present. That at the ends most likely defines the chance of resistance development. This information thus is crucial for the interpretation of this paragraph.

Subsection “Determinants of antibiotic resistance acquisition and microbiological treatment failure” – the argument may be valid, however, without data on the treatment regimen and the resistance data at a given timepoint, this is hard to made. This information needs to be provided.

Subsection “Antibiotic resistance and PE/PPE genes vary while antigens remain conserved” – variants in PE/PPE genes – in the opinion of this reviewer, a valid data analysis with regard to SNPs is impossible in repeat regions. Simply as you can not really allocate repeat block to a particular region in the reference genome. In addition, what are you doing with regions not present in the reference genome? If the authors want to keep this analyses in, they need to include a different way of confirmation e.g. Sanger sequencing of the regions from the strains of interest.

Subsection “Identifying candidate pathoadaptive loci from genome-wide variation” – resistance related genes need to be excluded here – or? That these have convergent evolution is not a surprise due to the selection pressure. This also induced the diversity. To really look into that, you would need to look into pan-susceptible isolates. What are the resistance types here?

Subsection “In-host mutations display phylogenetic convergence across multiple global lineages” – How do you distinguish site of likely sequencing error from real convergent evolution sites?

Discussion

– Please distinguish what is your finding and what is reported by others.

– “such patients”? What patients are you referring to? In your paper, you are not detailing patient characteristics. So, conclusions are difficult. In addition, I would be really cautious with the statement "relatively high percentage" develop resistance. To assay that, you would need to have a well-defined clinical trial in which you have clear cut in depth data for the patients characteristics incl. treatment data for the course of the treatment in relation to fixed and emerging resistances. At best with more than two serial isolates (see also comments above).

– Please reflect a bit more about the status of the published research here. What have others reported and detected? There have been several case reports already looking into that and other studies using new bioinformatic tools are available also.

Reviewer #2:

The manuscript by Vargas et al., presents a re-analysis of longitudinal samples for 307 TB patients. The analysis allows the authors to infer patterns of genetic diversity within a patient over time and the impact of different selective forces including treatment and host-pathogen interactions. The manuscript documents very well the analysis done as well as the different steps to reach the conclusions. However, in my opinion it lacks clarity and to some point novelty as it identified mostly known targets of drug resistance or targets of H-P difficult to validate. The methodology is however novel and relevant showing that in-host variation can be robustly analyzed in MTBC. I have some comments I would like the authors to address:

One major problem is that the authors lack associated clinical, demographic and epidemiological information about the cases under study. This is a major limitations as the authors can only look at the impact of bacterial genetic background (as infer from the WGS analysis) but not at relevant factor that we know or we suspect can influence levels of genetic diversity: HIV status, diabetes, treatment regimen, clinical adherence, nature of the lesions….etc. The authors correctly pool all the cases as failure (delayed culture conversion, treatment failure, relapse) but the reasons behind can be many and can impact the interpretation of some results. For example, H-P interaction loci will likely be linked to HIV status.

Subsection “Identifying clonal Mtb populations in-host”. The 7 SNPs threshold to discard a clonal infection seems misleading. Such an amount of variants could be compatible with clonal diversification in some cases with longitudinal samples taken several months apart and under antibiotic treatment pressure. From Figure 1C we can see the first reinfection case would have a 19-SNP difference, which seems compatible with the presence of two genotypes. But how would you classify a patient with a 10-SNP difference if you had it?

Figure 3. Related with the above question. One there is selection for a DR allele you expect a sweep of a particular clone, in practice this should translate in a decrease of diversity from Sample 1 to sample 2. Can you see this effect in Figure 3? Can you mark values for cases in which selection of DR is involved? Even more, in other patients where selection for H-P interaction is likely going on, can you see the effect? This will reinforce the idea that indeed those loci are involved in positive selection

Subsection “In-host pathogen dynamics in antibiotic resistance loci”. The 5% change threshold seems low if the variant is already at low frequency in the first sample. Of course, this is heavily influenced by read depth, but e.g. a change from 4% to 9% may be contributed by as little as 3 reads at 60X. The value could be adjusted dynamically according to the read depth stat for every sample.

Subsection “In-host pathogen dynamics in antibiotic resistance loci”. Wouldn't we expect a slightly higher mutation rate in this type of patients enriched with resistant strains and who failed treatment? On one hand because DR varaints are being fixed but also because It is expected to have some kind of hitchhiking effect during positive selection of DR variants in a clonal population. How the rate varies patient by patient?

Subsection “Simulations and PacBio sequencing demonstrate a low false-positive rate in repetitive regions”. Do the frequencies of SNPs in PE/PPE genes correlate in PacBio vs Illumina results? Also, are they mostly fixed or variable? A scatter plot maybe good here. This has implications to discuss about adaptation to host and the rate at which that would happen. In general, there is little information about the pacbio analysis and if it validates not just PE/PPE variation but other variation described for the patients sequenced with both technologies.

Discussion. Any reference to compare the presented value given the varied sources of the samples?

Discussion. If we talk about reinfections, not only is routinely performed sequencing advisable as the authors suggest, but also on subsequent samples from the patient to identify the second strain and adjust the treatment if needed.

Discussion. The low sensitivity of the 19% frequency threshold could be explained by the fact that random mutations appearing along a genotype that has acquired a drug resistance SNP (or any other variant that increases its fitness) will get fixated even if they have no phenotypic effect. How many synonymous mutations fall in this category? If you narrow down only to nonsynonymous will you increase sensitivity?

Subsection “In-host mutations display phylogenetic convergence across multiple global lineages”. In-host diversity. The authors analyze the gene-bygen diversity in-host. However, selection in-host does not necessarily reflect epidemiological success. The authors have the chance to look at it by comparing diversity within the host versus diversity between hosts (not just with the serial sample dataset but comparing to the reference collection of global isolates). Is there a correlation between in-host diversity vs between-host diversity?

Reviewer #3:

In this manuscript, the authors compile a significant body of work analyzing in-host population dynamics of *Mycobacterium tuberculosis*. The authors appropriately make use of publicly available data to compile a large dataset of paired samples in the same study participants over time, and the laboratory, bioinformatic, and statistical methods employed were well-designed to answer the questions of this manuscript. Overall, the analysis of 200 study participants from 8 studies has several important findings, including the low frequency of new resistance mutations in these participants, the importance of heteroresistance, in which minority variants representing {greater than or equal to}19% of reads predicted fixation in future samples, the significant contribution of prior resistance to development of new resistance, the greater role of drug resistance-associated mutations developing during drug treatment rather than new epitope-related mutations, confirmation of the development of new mutations in samples representing globally diverse lineages, and confirmation of a mutation rate within these samples that matches that of previous studies. Their findings suggest an important contribution of WGS to the prediction of treatment failure due to the potential superiority of WGS over phenotypic and rapid testing methods to identify heteroresistance at the start of treatment, which could presumably reduce the risk of treatment failure by indicating the need to adjust treatment regimens early.

While this manuscript represents a significant contribution to the field, a few considerations ought to be addressed. First, the use of public data, while laudable and appropriate for the aims of this study, introduces significant heterogeneity in the timing of sample collection and specific treatment regimens received. This does not, on its own, negatively affect the work of the manuscript, but the extent to which the authors combine these varied treatments and sample collection time points into a discussion of treatment failure and the relative contributions of resistance-associated mutations vs selection from the host's innate immune system requires further discussion. Supplementary file 1 summarized the heterogeneity of treatment regimens and sample collection time points to the extent that they are available. This indicates that some samples were collected before, during, and after treatment. Similarly, participants received diverse combinations of isoniazid, rifampin, rifapentine, pyrazinamide, ethambutol, streptomycin, and moxifloxacin with randomization of participants within each study introducing significant heterogeneity of selection pressure and time frames across samples in this study. The authors note a mutation frequency similar to the range derived in the absence of drug pressure (subsection “Characteristics of mutations in-host”) and refer in the Discussion section to inadequate therapy, which is hard to interpret with such heterogeneity. Due to the impact of baseline resistance (defined as allelic frequency >75%) and MDR disease on the development of new mutations during treatment, greater discussion of individual sample timing and duration/type of drug pressure would be helpful, as would be discussion of the allelic frequency threshold used to define prior resistance. Figure 6 confirms the significant impact of drug pressure as the primary driver of these mutations, rather than mutations in epitope encoding genes, so the extent to which mutations in epitope encoding genes are specifically varying in response to host activity (or not varying) is not clearly related to the host response from the data as presented. Similarly, the authors note that mutations in drug resistance-associated loci is common and occurs across distinct Mtb lineages. While the finding of phylogenetic convergence is important, an alternative framing of the finding would be to consider these sites to vary in the presence of drug pressure independent of lineage.

Of related concern, the authors defined study participants as having met criteria for failed therapy due to culture positivity after only 2 months of treatment. While patients receiving appropriate treatment ideally develop early culture conversion, many sources would require a longer time frame than 2 months to assign treatment failure, particularly if these participants were receiving experimental study regimens. Due to the heterogeneity introduced by the sample inclusion strategy, the conclusions of the manuscript with respect to negative treatment outcomes should be presented with greater weight placed on time between samples and time since starting therapy. Reframing the findings of the study as changes that occur during treatment, rather than changes occurring in the setting of failure, could also improve the support for the conclusions of the manuscript. For example, the authors identify a strong correlation between the SNP diversity of each participant's first sample and second sample and conclude that this demonstrates ineffective therapy (subsection “Determinants of antibiotic resistance acquisition and microbiological treatment failure”). How did this correlate among those participants who were not thought to have failed therapy? Did this vary with time to culture conversion among those converted later (that is, differentiating between those with "appropriate culture conversion", "delayed culture conversion," and true "failure"). A comparison between participants with eventual success and those who either never converted or who changed regimens due to emerging resistance would help differentiate this issue. Alternatively, a comparison between the SNP diversity among the 44 participants excluded due to different strains might better support the conclusion in the discussion that the sustained diversity identified is due to the absence of effective therapy.

Finally, the Introduction is a bit confusing, combining discussions of the impact of host immunity, drug pressure, and microbiological and sequencing biases in selection of bacterial subpopulations. The result is that the reader is left confused about how to frame their interpretation of the study findings. This may be improved by a simpler introduction of the problem of unknown in-host variation over time and a summary of experiments that explain which bacterial factors should be considered at baseline to help predict future changes in Mtb isolates as well as the fixation of mutations in different genomic loci over time.

---

## [Author Response]

[Editors’ note: The authors appealed the original decision. What follows is the authors’ response to the first round of review.]

Reviewer #1:The study aims to provide insights in the evolution of M. tuberculosis in the host with a particular focus on resistance development. The authors can show, that pre-existing mutations predict failure and can be fixed in follow up samples. That is per se not a new finding, but the data are comprehensive and provide a detailed view into in host dynamics in patients with failing regimens.(1.1) What I found problematic in the study is that the authors work with only two samples per patient, thus, not really allowing the analyses of the longitudinal evolutionary dynamics in TB patients. Accordingly, the title should be modified to reflect more the work the authors performed.

We thank the reviewer for raising these two points. We agree that other smaller studies have attempted to study longitudinal isolates, largely with the focus of distinguishing re-infection with a new *M. Tuberculosis* (Mtb) strain from true treatment failure *i.e.* continued infection with the same strain. We want to emphasize that no other study to our knowledge has investigated Mtb in-host evolution at the scale of this study. Given the low rate of TB failure and relapse after treatment of drug susceptible TB, usually <10%, and moderate rate for drug resistant TB, ~30-40%, as well as Mtb’s slow rate of evolution, it is necessary to pool data from many studies to explore this question with sufficient statistical power. This is precisely what we do in this study. Although we do not study multiple longitudinal samples per patient, and focus our analysis on two samples per patient, we believe this constitutes a significant advance of knowledge. This is especially the case as we largely focus on large allele frequency changes between the first and second time point and do not study transient or very low frequency variation that may require sampling at more than two time points. The two samples we study are longitudinally collected from each patient with dates of collection documented for nearly all isolates. In the Results section (Genome-wide in-host diversity), we describe how we used technical replicates to approximate the noise that arises from sampling and how a large change in allele frequencies between sampling is likely indicative of a mutant allele getting purged or sweeping to fixation in-host. This benchmarking constitutes an important methodological advance, in our opinion, for studying pathogen in-host evolution.

We have now modified the title to read: “In-host population dynamics of *M. tuberculosiscomplex* during active disease” to de-emphasize the study of strict treatment failure and address the reviewer’s comments below. We believe this title accurately reflects the contents of the manuscript and welcome additional suggestions for edits.

(1.2) Also, the selection of the resistance variants during therapy has been shown several times in previous papers and is the natural process for resistance development. So, the information presented is not really new.

We agree with the reviewer that the selection of resistance variants during therapy has been demonstrated previously. This was not the main finding of our study. Our goal in reporting this result was to extend previous analyses by providing a quantification of the rate of resistance acquisition. We also sought to provide a comparison of the rate of resistance amplification between drug resistant and drug susceptible isolates, and most importantly to study allele frequency dynamics. Specifically, the analysis where we determine the allele frequency above which we observed consistent fixation of the drug resistant allele, is novel and has not been previously reported. We also demonstrate a pattern of clonal interference for sites in resistance genes not known to be canonical resistance variants e.g. in Figure 2A *katG* L159F. Lastly and beyond drug resistance acquisition we study most of the Mtb genome (i.e., not just loci implicated in antibiotic resistance) to investigate selection for variants unrelated to antibiotic treatment. We believe these analyses to be novel and expand our knowledge of Mtb evolution.

(1.3) What I find interesting is the calculation of the "Lowest" low frequency level that can be detected. However, the paragraph is not really conclusive and the authors should refer to recently published papers e.g. Dreyer et al., 2020.

We thank the reviewer for pointing out the paper by Dreyer et al., 2020, we have now cited this reference in the paragraph describing the lowest frequency level. We have also revised this paragraph to add clarity to the results. In particular, we note that we are specifically interested in the “lowest” low frequency level that can be detected at *the first time point* and predict the fixation of that mutant allele at *the second time* point. This is to be distinguished from the ability to detect very low allele frequencies (~1%) accurately at a single time point which is what Dreyer et al., focus on by validating against simulated sequence data and in vitro mixtures. Notably Dreyer et al., report false positive calls only at allele frequencies lower than 5% in their study. All analyses in our paper focus on alleles at a frequency of >5%, requiring minimum depth of 25x, and read depth of at least 5 reads supporting the alternate allele. The average depth of read coverage in our study isolates was 186x. Hence the Dreyer et al. results strongly support our choice of allele frequency thresholds and depth as having a very low risk of SNPs called due to sequence error alone.

(1.4) Overall, the paper is well written, however, more data esp. patient data need to be provided to allow a clear cut interpretation of the data presented.

We thank the reviewer for describing the manuscript as such, we aimed to communicate the results with accuracy and clarity. In response to the reviewers’ requests for additional patient data, we have now aggregated treatment from the source studies and added this metadata to Supplementary file 2 for each longitudinal isolate. We include columns that indicate the timing of sampling of Mtb relative to treatment, the treatment regimen administered and final patient outcome (and relevant details). Patient outcomes are defined as follows: Delayed culture conversion (sputum culture positive at baseline and ≥ 2 months treatment initiation with genomic analysis consistent with clonal infection), Failure or Relapse (sputum culture positive at baseline and ≥ 4.5 months treatment initiation with genomic analysis consistent with clonal infection), Failure or Relapse or Default (sputum culture positive at interval of ≥ 4.5 months with genomic analysis consistent with clonal infection, only partial treatment data is available) or N/A if date data was of low resolution, not available or no treatment data was available. We also determined Reinfection and Mixed infection based on the genomic analysis. This information has been added to the Materials and methods section.

We have now also repeated our analyses on the allele frequency dynamics within antibiotic resistance (AR) loci and rates of resistance amplification using the subset of 121/200 patients with confirmed failure or relapse (Supplementary file 2) corresponding to the following sections:

Results: In-host pathogen dynamics in antibiotic resistance loci, Allele frequency >19% predicts subsequent fixation of resistance variants, Determinants of antibiotic resistance acquisition and microbiological treatment failure). For all 200 cases the order of sampling was available, but for 195/200 (119/121 confirmed failure subjects) we also had the exact dates of sampling which were required for some analyses. We found that the analysis conclusions were unchanged between both the 121 subset and the full 200 patient sample. We have now added the results of these analyses in the supplementary text as Appendix 3.

(1.5) Introduction - I would refer to M. tuberculosis complex

The reviewer is correct, as we have included Lineages 5 and 6, we have now revised “*Mycobacterium tuberculosis”* to *“Mycobacterium tuberculosis* complex*”* in the title and Introduction.

Results(1.6) Subsection “Identifying clonal Mtb populations in-host”- where are the data concerning MTB lineage shown?

The data are shown in Figure 5A. We have now added a reference to this figure in the text.

(1.7) Subsection “In-host pathogen dynamics in antibiotic resistance loci” – how can you differentiate contaminations from real low frequency variants if you have only the two datapoints? What statistical method did you use to distinguish sequencing errors from real SNPs?

The analysis involved multiple steps of quality control including control for contamination, steps to differentiate sequencing error from real low frequency SNPs at each time point and steps to determine the significance of allele frequency changes between time points. Contamination of sequencing read data was evaluated for each isolate and does not require comparison of data at different time points. We used a highly cited method for contamination detection, Kraken, (PMID: 24580807), that was recently also validated for use in Mtb (PMID: 32122347). Patients with one or more isolates without > 95% of reads mapping to Mtb complex were excluded as contaminated. We required a minimum read depth of 25x at each time point and that the mutant allele frequency changes by at least 5% between paired (longitudinal) isolates for us to include the drug resistance SNP in this analysis. To ensure a minimum of 5 reads supports each mutant allele called (with min. read depth of 25x), we also required an additional filter if the change in allele frequencies fell between 5% and 20%: these SNPs were only retained if the mutant allele (in both isolates) that had an allele frequency > 0% was supported by at least 5 reads at either time point. As we trimmed all reads to a phred error of 1/100, the depth and allele frequency threshold associated with a probability of sequencing error of (1/100)^(5)=1x10^-10^. Furthermore, we studied alleles changing in allele frequency by >5% only in drug resistance genes as they are known to be under selection. For other genes in the genome, we only count SNPs that have increased in allele frequency by > 70% as significant for *in host* evolution based on our comparison with serial paired samples that were only passaged in vitro. We found that smaller allele frequency changes may be observed simply because of in vitro passage. These methods are detailed in the Materials and methods section.

To aid the readers, we have added the following descriptions of our methodology directly into the Results:

Subsection “Identifying clonal Mtb populations in-host”: “We required that no indels be present in the pileup supporting any SNP call, dropped SNP calls in repetitive regions and enforced a read depth ≥ 25x and alternate allele depth of ≥5 reads.”

Subsection “In-host pathogen dynamics in antibiotic resistance loci”: “To investigate temporal dynamics related to antibiotic pressure, we identified non-synonymous and intergenic SNPs within a set of 36 predetermined resistance loci associated with antibiotic resistance (Farhat et al., 2016, 2013) (Supplementary file 5 ) that changed in allele frequency by ≥5% (Sun et al., 2012) and ensuring that support of the alternate allele was ≥5 reads at each time point (Materials and methods ).”

(1.8) Subsection “Allele frequency >19% predicts subsequent fixation of resistance variants” – this paragraph is not really clear to me – what was the lowest frequency then that can be detected and results in treatment failure? Please look at Dreyer et al., 2020.

We apologize about this lack of clarity. The goal of this analysis is to find the lowest mutant allele frequency that can be detected at the first time point and can be used to predict the fixation of the mutant allele in the sample taken at the second time point. The majority of our isolates were collected from patients with treatment failure, and we did not have a comparator group of patients cured with treatment to causally link resistance allele frequency with treatment failure. We do not claim to do this in the text, and we believe this association between allele frequency with the probability of fixation lays the foundation for future studies that can causally link to treatment failure. We identified this lowest allele frequency at the first time point to be 19%.

In response to the reviewer’s comment and other comments below, we have now simplified the paragraph to add clarity as follows:

“We aimed to measure the lowest AR allele frequency that can accurately predict the fixation of resistance alleles later in time (Dreyer et al., 2020; Sun et al., 2012; Zhang et al., 2016). We examined all 1,919 SNPs that varied by at least 5% in allele frequency (AF), and discarded 20 SNPs that were fixed at AF > 75% in both isolates. We calculated the true positive rate (TPR) and false positive rate (FPR) for varying values of AF at the first time point (AF1) ∈{0,1,2,…,99,100}% (Figure 2C, Materials and methods) allowing a maximum FPR of 5%. We found the optimal classification threshold to be with an associated sensitivity of 27.0% and a specificity of 95.8%.”

(1.9) Subsection “Determinants of antibiotic resistance acquisition and microbiological treatment failure” – don't understand this sentence – If the patients have treatment failure, they are treated all the time – or?

We apologize about this lack of clarity. We have now added treatment information into Supplementary file 2 as detailed above. We have also rephrased this sentence to make it clearer and pointed readers to the supplementary table that has the treatment metadata. The sentence currently reads:

“Among the set of 195/200 patients with clonal infection and sampling date, AR acquisition was more likely as the time between sampling increased with the OR of AR acquisition being 1.023 per 30day increment (95% CI 1.002, 1.045, P=0.035 Logistic Regression).”

(1.10) Subsection “Determinants of antibiotic resistance acquisition and microbiological treatment failure” – the finding that pre-existing resistance mutations are leading to a higher chance of failure and resistance development is not surprising. What needs to be provided here are data on the effectivity of the given regimen. Meaning, one needs to know with how many active drugs a patient was treated when the low level resistance mutation was present. That at the ends most likely defines the chance of resistance development. This information thus is crucial for the interpretation of this paragraph.

We thank the reviewer for asking for additional clarity on this point. In Supplementary file 2 we now add detailed treatment data, and classify outcomes as detailed above including “delayed culture conversion” or “treatment failure and relapse”. These outcomes were defined by persistent positive cultures on treatment detected beyond 2 months and 4.5 months from treatment initiation. Detailed drugs administered are also given in Supplementary file 2, Drug resistance profiles are also given in Supplementary file 21. We have now repeated this analysis focusing specifically on patients with confirmed failure i.e. excluding patients where treatment details were not available or for which we could not exclude default. We explored the specific question the reviewer raised about the number of effective drugs received by the 11 patients that demonstrated evidence for resistance acquisition. We found that in 9/11 cases the patient received fewer than four effective drugs. This result has now been added to the Subsection “Determinants of antibiotic resistance acquisition and microbiological treatment failure”.

(1.11) Subsection “Determinants of antibiotic resistance acquisition and microbiological treatment failure” – the argument may be valid, however, without data on the treatment regimen and the resistance data at a given timepoint, this is hard to made. This information needs to be provided.

For ease of reference here is the argument the reviewer is referring to – “We also quantified genome-wide Mtb diversity in-host among the patients with microbiological treatment failure….”

We have now aggregated patient treatment from the source studies and added this metadata to Supplementary file 2 for each longitudinal isolate (see above). We include columns that indicate the timing of sampling of Mtb relative to treatment, the treatment regimen administered and final patient outcome (and relevant details). We believe this updated treatment regimen and resistance data better support the claim made in the text. We have now also revised the text to read:

“We also quantified genome-wide Mtb diversity in-host among the patients with persistent or relapsed infection for ≥2 months. We reasoned that if these patients are not on or not adherent to effective antibiotic treatment, their effective pathogen population size may be large and prone to more genetic drift or turnover of minority variants with and without selection (Trauner et al., 2017).”

(1.12) Subsection “Antibiotic resistance and PE/PPE genes vary while antigens remain conserved” – variants in PE/PPE genes – in the opinion of this reviewer, a valid data analysis with regard to SNPs is impossible in repeat regions. Simply as you cannot really allocate repeat block to a particular region in the reference genome. In addition, what are you doing with regions not present in the reference genome? If the authors want to keep this analyses in, they need to include a different way of confirmation e.g. Sanger sequencing of the regions from the strains of interest.

The reviewer is correct to raise concern about variant calling in repetitive regions of the Mtb genome. We want to emphasize that while the PE/PPE genes are usually excluded from genomic analyses, a large proportion of these regions are not repetitive and allow for variant calling with high accuracy. Many PE/PPE genes which do contain repetitive sequence content also contain uniquely alignable sequences. We have extensively validated that the variant calls made in our study are not due to mapping or sequencing error in the following three ways:

1) We used extensive filters for calling all SNPs (see Materials and methods). (2) We simulated all the SNPs that we observed in-host (including the ones detected in the PE/PPE regions) in a diverse set of complete genomes, simulated short-read sequencing data from these genomes then mapped the simulated reads to H37Rv and called SNPs with our pipeline. These simulations demonstrated that we could reliably call the SNPs we observed in-host from short-read sequencing data (see subsection “Simulations and PacBio sequencing demonstrate a low false-positive rate in repetitive regions”, subsection “SNP Calling simulations in repetitive genomic regions” and Appendix 1). Finally, (3) We used paired PacBio and Illumina sequences (taken from the same isolates) and compared the congruence of base calls across the genome using the PacBio sequences as a “ground truth” since PacBio reads are much longer and are used routinely to sequence repetitive regions. After comparing the congruence of calls between both sequencing technologies, we only called SNPs in regions of the genome where the base calls from Illumina agreed with the PacBio base calls (see subsection “Simulations and PacBio sequencing demonstrate a low false-positive rate in repetitive regions”, subsection “Empirical Score for Difficult-to-Call Regions” and Appendix 2).

(1.13) Subsection “Identifying candidate pathoadaptive loci from genome-wide variation” – resistance related genes need to be excluded here – or? That these have convergent evolution is not a surprise due to the selection pressure. This also induced the diversity. To really look into that, you would need to look into pan-susceptible isolates. What are the resistance types here?

We agree with the reviewer that convergent evolution occurs within the antibiotic resistance genes due to selection pressure. We included the drug resistance loci as a “Positive control” or comparator set of genes. Given that we hypothesize that other selection pressures (e.g., immune-related) might account for some of the other diversity, the observation of a strong signal in these resistance genes is a good sign that our methodology works and provides a point of reference of the strength of the selection signal. We break down the SNP frequencies that we detect in-host into different categories: Essential, Non-Essential, Antigen, PE/PPE and Antibiotic Resistance genes so we separate the diversity stemming from antibiotic pressure from the diversity likely arising from other selection pressure or genetic drift. See Figure 6 C-D in the main text to demonstrate this.

(1.14) Subsection “In-host mutations display phylogenetic convergence across multiple global lineages” – How do you distinguish site of likely sequencing error from real convergent evolution sites?

For variant calling in the globally representative 20,352 isolates we used conservative variant calling filters to minimize the effects of sequencing error. Briefly, for each call, we required mean base quality > 20, mean mapping quality > 30, excluded SNPs called by reads also supporting an insertion or deletion, a minimum depth of 25 reads (Materials and methods). Additionally, we restricted our analysis to regions of the genome that were confirmed callable with high accuracy when compared with PacBio sequencing data (Subsection “Empirical Score for Difficult-to-Call Regions”) and excluded positions located within mobile genetic elements (subsection “Variant Calling/SNP Calling”). We also required that the allele called for a given SNP site be pure *i.e.* supported by at least 90% of the reads (Materials and methods ) otherwise we marked a missing call at that SNP site in that isolate. Finally, we also dropped SNP sites that had missing calls in > 25% of isolates and then dropped isolates that had missing calls > 25% of SNP sites to further exclude low-quality SNP sites and low-quality sequenced isolates (Materials and methods). Taken together, we believe this extensive filtering removed sites prone to sequencing error.

Discussion(1.15) Please distinguish what is your finding and what is reported by others.

We have rearranged this paragraph to aid the reader in distinguishing our findings (later in the paragraph) from what has been reported by others (moved to earlier in the paragraph).

“In our Mtb populations sequenced from active TB patients enriched for negative treatment outcomes, we find a wealth of dynamics in genetic loci associated with antibiotic resistance, including a high turnover of minor variants. Known factors that determine treatment outcome are complex and include severity of lung disease, cavitation and adherence to treatment among others (Imperial et al., 2018). Additionally, resistance acquisition in the course of one infection is comparatively rare in most pathogenic bacteria (Llewelyn et al., 2017). Here, we observe that 9% of patients with confirmed delayed culture conversion, failure and relapse amplify resistance over time. Our findings of a higher rate of resistance acquisition in patients with MDR at the outset and with time between sampling, emphasize the importance of appropriately tailoring treatment regimens as well as close surveillance for microbiological clearance and resistance acquisition by phenotypic or genotypic means. The observed high rate of resistance acquisition also emphasizes Mtb’s biological adaptability and the long duration of drug pressure in vivo. In addition to clonal acquisition of resistance, we find that sequencing revealed a substantial proportion of mixed infection or reinfection (28% of samples collected ≥2months apart). This high percentage suggests that patient treatment and control of disease transmission can be better guided if pathogen sequencing is routinely performed for cases with persistent positive cultures especially in high TB prevalence settings where reinfection is more likely. Reinfection can also introduce strains with a different antibiotic susceptibility profile requiring adjustment in the treatment regimen.”

(1.16) “such patients”? What patients are you referring to? In your paper, you are not detailing patient characteristics. So, conclusions are difficult. In addition, I would be really cautious with the statement "relatively high percentage" develop resistance. To assay that, you would need to have a well defined clinical trial in which you have clear cut in depth data for the patients characteristics incl. treatment data for the course of the treatment in relation to fixed and emerging resistances. At best with more than two serial isolates (see also comments above).

We have modified this calculation to include only the 119 subjects that we have confirmed delayed culture conversion, failure or relapse, have appropriate isolate date collection data for both timepoints and have detailed treatment data (Supplementary file 2).

“Here, we observe that 9% of patients with confirmed delayed culture conversion, failure and relapse amplify resistance over time.”

(1.17) Please reflect a bit more about the status of the published research here. What have others reported and detected? There have been several case reports already looking into that and other studies using new bioinformatic tools are available also.

In response to the reviewers comment, we have now added a few sentences to add background on what others have reported and clarity to this paragraph. This additional information relates our study to similar research questions tackled by other studies and cited (PMID: 28424085) and (PMID: 32398743) accordingly.

“While prior studies have investigated the lowest resistance allele frequency that can be detected in clinical sputum samples (Dreyer et al., 2020; Trauner et al., 2017), there is little information on the clinical relevance of these low frequency variants. We provide a proof-of-concept analysis that minor AR alleles, occurring at a frequency ≥19%, can predict fixation of the variant with a specificity >95% of mutations in-host, although we find the sensitivity of this threshold to be low. The low sensitivity is because the majority of alleles that sweep to fixation are actually not detectable at all at the first time point, suggesting that more frequent sampling may be needed. In the future, higher depth and more frequent sequencing can elucidate more clearly the role of minor AR allele detection in clinical management of TB treatment.”

Reviewer #2:The manuscript by Vargas et al., presents a re-analysis of longitudinal samples for 307 TB patients. The analysis allows the authors to infer patterns of genetic diversity within a patient over time and the impact of different selective forces including treatment and host-pathogen interactions. The manuscript documents very well the analysis done as well as the different steps to reach the conclusions. However, in my opinion it lacks clarity and to some point novelty as it identified mostly known targets of drug resistance or targets of H-P difficult to validate. The methodology is however novel and relevant showing that in-host variation can be robustly analyzed in MTBC. I have some minor and major comments I would like the authors to address:(2.1) One major problem is that the authors lack associated clinical, demographic and epidemiological information about the cases under study. This is a major limitations as the authors can only look at the impact of bacterial genetic background (as infer from the WGS analysis) but not at relevant factor that we know or we suspect can influence levels of genetic diversity: HIV status, diabetes, treatment regimen, clinical adherence, nature of the lesions….etc.The authors correctly pool all the cases as failure (delayed culture conversion, treatment failure, relapse) but the reasons behind can be many and can impact the interpretation of some results. For example, H-P interaction loci will likely be linked to HIV status.

We thank the reviewer for raising this concern. It is similar to concerns raised by reviewer 1 to which we provide detailed response and describe edits to the manuscript above in Response 1.4. We believe strongly that the manuscript is now significantly improved as a result of this feedback. There are some metadata patient characteristics that we were unable to locate and aggregate such as HIV status, diabetes and nature of the lesions. We agree with the reviewer that such host characteristics (e.g., HIV) may alter selective pressures on the pathogen during active disease. Future studies examining the association between HIV and convergent evolution in genes identified here to be associated with H-P (host-pathogen) interactions will be worthwhile. We note that a large proportion of isolates included in this study derive from low HIV incidence settings specifically: Peru, the United Kingdom and China that are the source sites for 144/307 patients. Additionally, although some isolates from Witney et al., and Bryant et al., 163/307 were isolated from patients with HIV, all enrolled HIV patients had to have a CD4 count >250 in those two studies. Hence, we expect the overall proportion of patients with HIV/AIDS to be low in our dataset. Though we lack this specific metadata, we believe that our validation of results across a large dataset of WGS data (~20k samples) constitutes an important first step and adds a new research direction for studying H-P relationships using convergent evolution. We demonstrate that in addition to loci involved the acquisition of antibiotic resistance loci implicated in modulation of innate host-immunity appear to be under positive selection.

(2.2) Subsection “Identifying clonal Mtb populations in-host”. The 7 SNPs threshold to discard a clonal infection seems misleading. Such an amount of variants could be compatible with clonal diversification in some cases with longitudinal samples taken several months apart and under antibiotic treatment pressure. From Figure 1C we can see the first reinfection case would have a 19-SNP difference, which seems compatible with the presence of two genotypes. But how would you classify a patient with a 10-SNP difference if you had it?

The reviewer is correct, the minimum SNP difference that we see among the reinfection cases is 19 SNPs and the largest SNP difference that we see among clonal infection is 7 SNPs. The 44 reinfection cases had a median SNP distance = 708 with IQR = [250.5, 1086.5] showing a large separation between reinfection cases and clonal infection cases. As such, our effective threshold for reinfection amounts to > 18 SNPs and our threshold for clonal cases was < 7 SNPs in our data. We agree with the reviewer that a pair of isolates 10-SNPs apart is likely consistent with clonal infection. The prior literature on the subject reports the use of a 12-SNP threshold suitable for epidemiological links via genomic sequencing (PMID: 23158499, cited 667 times). We did not have any such borderline cases in our dataset. In response to the reviewer’s comment, we have now added the median SNP distance and IQR range for the reinfection cases to the main text to communicate than none of these cases were borderline.

(2.3) Figure 3. Related with the above question. One there is selection for a DR allele you expect a sweep of a particular clone, in practice this should translate in a decrease of diversity from Sample 1 to sample 2. Can you see this effect in Figure 3? Can you mark values for cases in which selection of DR is involved? Even more, in other patients where selection for H-P interaction is likely going on, can you see the effect? This will reinforce the idea that indeed those loci are involved in positive selection

We thank the reviewer for their suggestion. We have modified Figure 3A in the following way (Figure 3 legend): “The number of hSNPs called in the second sample isolated vs the number of hSNPs called in the first sample isolated from each of 178 subjects (median T1=13.5 hSNPs, median T2=13.5 hSNPs). The dashed line is y = x. Red denotes 27/178 patients who had an antibiotic resistance in-host SNP arise between sampling (median T1=15.0 hSNPs, median T2=11.0 hSNPs), blue denotes 5/178 patients who had a putative host-adaptive in-host SNP (Rv1944c, Rv0095c, PPE18, PPE54, PPE60) arise between sampling (median T1=19.0 hSNPs, median T2=6.0 hSNPs).” We observe a lower median number of hSNPs at the second time point for the subjects in which DR alleles sweep to fixation and in which putative H-P alleles sweep to fixation, demonstrating a reduction in diversity and reinforcing the idea that these loci may be involved with positive selection. We have added this observation to the second-to-last paragraph of the Discussion: “Consistent with the idea that positive selection is acting on alleles within these loci, we observe a reduction in diversity at the second time point for the subjects in which drug resistant alleles sweep to fixation and in which putative host-pathogen alleles sweep to fixation (Figure 3A).”

(2.4) Subsection “In-host pathogen dynamics in antibiotic resistance loci”. The 5% change threshold seems low if the variant is already at low frequency in the first sample. Of course, this is heavily influenced by read depth, but e.g. a change from 4% to 9% may be contributed by as little as 3 reads at 60X. The value could be adjusted dynamically according to the read depth stat for every sample.

We thank the reviewer for raising this concern. It is similar to two concerns raised by reviewer 1. We refer to the reviewer to a detailed description of depth and other quality control criteria and their discussion in relation to a recent publication on the topic in Mtb by Dreyer et al., under Response 1.3 and 1.7 above. Under these two responses we also list detailed edits to the text to add clarity. We note that we only looked at small allele frequency changes, down to 5%, in drug resistance genes as they are known to be under selection. In other genes in the genome, we only count SNPs that have increased in allele frequency by > 70% as significant for in host evolution based on our comparison with serial paired samples that were only evolved in vitro.

(2.5) Subsection “In-host pathogen dynamics in antibiotic resistance loci”. Wouldn't we expect a slightly higher mutation rate in this type of patients enriched with resistant strains and who failed treatment? On one hand because DR varaints are being fixed but also because It is expected to have some kind of hitchhiking effect during positive selection of DR variants in a clonal population. How the rate varies patient by patient?

We thank the reviewer for this suggestion. In Figure 6B, we observe that fixed

SNPs accumulate at an average rate of 0.56 SNPs per genome per year (95% CI = [0.408, 0.708], *P* = 7x10^-12^) when regressing the number of SNPs per subject on the timing between isolate collection for 195/200 subjects with isolate collection dates. Per the suggestion above, we also regressed the number of SNPs per subject on the timing between collection for 119 confirmed failure/relapse subjects. In this subset of patients, we observe that SNPs accumulate at an average rate of 0.64 SNPs per genome per year (95% CI = [0.432, 0.84], *P* = 7x10^-9^). Although this rate is marginally higher, the 95% confidence intervals overlap substantially. The observed number of new mutations is expectedly stochastic and varied considerably from patient to patient with only 71/200 subjects developing ≥1 in-host SNP and the maximum number of in-host SNPs observed being 5. The analysis of the mutation rate in the 119-patient subset has now been added to the supplement along with a supporting figure and table (Appendix 3—figure 3, Supplementary file 2).

(2.6) Subsection “Simulations and PacBio sequencing demonstrate a low false-positive rate in repetitive regions”. Do the frequencies of SNPs in PE/PPE genes correlate in PacBio vs Illumina results? Also, are they mostly fixed or variable? A scatter plot maybe good here. This has implications to discuss about adaptation to host and the rate at which that would happen. In general, there is little information about the pacbio analysis and if it validates not just PE/PPE variation but other variation described for the patients sequenced with both technologies.

Calling low allele frequency variants in microbial genomes using PacBio sequencing data does not yet have an established methodology. This is in part due to PacBio reads having much lower per base accuracy compared with Illumina; for our PacBio data, reads have an estimated per base error rate of 10%. We note that the study of within sample diversity with PacBio sequencing is an active area of research, and we hope that new methods will be developed to use PacBio data in this space in the near future. For this study, we did use paired PacBio and Illumina sequences (taken from the same isolates) and compared the congruence of base calls at high allele frequencies (>75%) across the genome. For this analysis we used PacBio based assemblies as a “ground truth” since PacBio reads are much longer and are used routinely to sequence repetitive regions. (see subsection “Simulations and PacBio sequencing demonstrate a low false-positive rate in repetitive regions”, subsection “Empirical score for difficult-to-call regions” and Appendix 2).

We have added the following sentence for clarity: “While the high per base error rate makes it difficult to call low allele frequency variants in microbial genomes, we made use of the PacBio sequencing data to assess fixed variant calls.” in the context of the following paragraph: “Second, we assessed the congruence in variant calls between short-read Illumina data and long-read PacBio data for a set of isolates that underwent sequencing with both technologies (Materials and methods). Unlike Illumina generated reads, PacBio reads are much longer and have randomly distributed error profiles (Rhoads and Au, 2015). With high coverage, PacBio sequencing can reliably reconstruct full microbial genomes and identify SNPs in repetitive regions. While the high per base error rate makes it difficult to call low allele frequency variants in microbial genomes, we made use of the PacBio sequencing data to assess fixed variant calls. The comparison with PacBio assemblies confirmed empirically a low rate of false positive base calls in genomic regions where we observed in-host SNPs (Materials and methods).”

(2.7) Discussion. Any reference to compare the presented value given the varied sources of the samples?

To our knowledge our paper is the first to measure the rate of resistance acquisition at this scale using longitudinal whole genome sequencing in patients with documented treatment failure. We also show a significant association with pre-existing resistance. There is unfortunately limited context from the literature to add to this section, but we hope the revisions we have made with regards to supplying treatment regimen data, as well as the number of effective drugs (see Response 1.10) now improves the interpretation of this result.

(2.8) Discussion. If we talk about reinfections, not only is routinely performed sequencing advisable as the authors suggest, but also on subsequent samples from the patient to identify the second strain and adjust the treatment if needed.

The reviewer raises a great point which has been incorporated into the Discussion by adding the following sentence: “Reinfection can also introduce strains with a different antibiotic susceptibility profile requiring adjustment in the treatment regimen.”

(2.9) Discussion. The low sensitivity of the 19% frequency threshold could be explained by the fact that random mutations appearing along a genotype that has acquired a drug resistance SNP (or any other variant that increases its fitness) will get fixated even if they have no phenotypic effect. How many synonymous mutations fall in this category? If you narrow down only to nonsynoymous will you increase sensitivity?

The sensitivity of the 19% frequency threshold may be explained by the detection of several competing clones at low frequency (AF < 19%) (Figure 2A, Figure 2—figure supplement 1) early on in treatment and then the subsequent fixation of one of these clones at a later point in time. The sensitivity may also be explained by the observation that most antibiotic resistance alleles that fix in the second time point are undetectable in the first time point as we now note in the revised text and describe further under Response 1.8. Additionally, our methodology restricted this analysis to study only intergenic and non-synonymous mutations that are located within antibiotic resistance loci. This detail on our filtering process for calling the variants for this analysis can be found in the Materials and methods section. We have added several sentences to further explain our observations to the discussion.

Edits to the manuscript are described above under response 1.8 and response 1.17.

(2.10) Subsection “In-host mutations display phylogenetic convergence across multiple global lineages”. In-host diversity. The authors analyze the gene-bygen diversity in-host. However, selection in-host does not necessarily reflect epidemiological success. The authors have the chance to look at it by comparing diversity within the host versus diversity between hosts (not just with the serial sample dataset but comparing to the reference collection of global isolates). Is there a correlation between in-host diversity vs between-host diversity?

We thank the reviewer for raising this point. We agree that selection in host does not reflect epidemiological success. This is specifically why we conducted the analysis assessing phylogenetic convergence across 20,352 isolates. Specifically reasoning that pathoadaptive mutations observed to sweep to fixation in-host and not compromise pathogen transmissibility are likely to arise independently within other subjects and in separate geographic regions in a convergent manner (PMID: 23995135). This is demonstrated by the detection of convergent resistance mutations (Figure 7B-C, Figure 7—figure supplement 1, Figure 7—figure supplement 2) along with other hypothesized pathoadaptive mutations. We agree that analyses comparing in-host and between host diversity would be valuable, but these would have to rely on available linked in-host and transmission/outbreak data at a sufficient scale. This type of data is not available to us and would likely require large data prospectively collected for this purpose.

Reviewer #3:In this manuscript, the authors compile a significant body of work analyzing in-host population dynamics of Mycobacterium tuberculosis. The authors appropriately make use of publicly available data to compile a large dataset of paired samples in the same study participants over time, and the laboratory, bioinformatic, and statistical methods employed were well-designed to answer the questions of this manuscript. Overall, the analysis of 200 study participants from 8 studies has several important findings, including the low frequency of new resistance mutations in these participants, the importance of heteroresistance, in which minority variants representing {greater than or equal to}19% of reads predicted fixation in future samples, the significant contribution of prior resistance to development of new resistance, the greater role of drug resistance-associated mutations developing during drug treatment rather than new epitope-related mutations, confirmation of the development of new mutations in samples representing globally diverse lineages, and confirmation of a mutation rate within these samples that matches that of previous studies. Their findings suggest an important contribution of WGS to the prediction of treatment failure due to the potential superiority of WGS over phenotypic and rapid testing methods to identify heteroresistance at the start of treatment, which could presumably reduce the risk of treatment failure by indicating the need to adjust treatment regimens early.(3.1) While this manuscript represents a significant contribution to the field, a few considerations ought to be addressed. First, the use of public data, while laudable and appropriate for the aims of this study, introduces significant heterogeneity in the timing of sample collection and specific treatment regimens received. This does not, on its own, negatively affect the work of the manuscript, but the extent to which the authors combine these varied treatments and sample collection time points into a discussion of treatment failure and the relative contributions of resistance-associated mutations vs selection from the host's innate immune system requires further discussion. Supplementary file 1 summarized the heterogeneity of treatment regimens and sample collection time points to the extent that they are available. This indicates that some samples were collected before, during, and after treatment. Similarly, participants received diverse combinations of isoniazid, rifampin, rifapentine, pyrazinamide, ethambutol, streptomycin, and moxifloxacin with randomization of participants within each study introducing significant heterogeneity of selection pressure and time frames across samples in this study.

The reviewer raises several important points. We agree that the manuscript lacked detail on treatment outcomes and the text may have focused too heavily on the discussion of treatment failure that is challenged in interpretation by the meta-analysis design. We have taken these critiques to heart and made extensive revisions as follows: We have generated a new and detailed patient treatment metadata Table (Supplementary file 2) that details treatment regimens received and describes exactly when samples were collected relative to treatment initiation. We refer the referee to response 1.4 above for additional details on this meta-data. We confirmed 121 cases of treatment failure based on treatment regimen data and sampling times, and an additional 57 cases had limited treatment data and include failure/relapse or default/treatment interruption. In the grand majority of cases, n = 117/121, samples were collected at the start of treatment and in follow up when treatment failure or relapse was identified.

We replicated all analyses focused on treatment/drug resistance (Appendix 3, Appendix 3—figure 1, Appendix 3—figure 2) specifically in the 121 cases of confirmed treatment failure and provide these results in the supplement. We refer the referee to Response 1.4 in which we report the findings of these replicate analyses and compare them to the analyses reported in the manuscript on the larger sample.

We have also now revised the text to change the focus from treatment failure to persistent active disease:

– Changed title from “In-host population dynamics of *M. tuberculosis* during treatment failure” to “In-host population dynamics of *M. tuberculosis* during active disease”

– Changed sentence from “Of the 178/200 subjects with unsuccessful treatment outcome” to “Of the 178/200 subjects with persistent clonal infection > 2 months” in Abstract

– Deleted the statement “All study subjects had either recently completed treatment or were receiving treatment when samples were collected…” from Results.

– Changed sentence from “In our Mtb populations sequenced from active TB patients enriched for delayed culture conversion, treatment failure and relapse…” to “In our Mtb populations sequenced from active TB patients enriched for negative treatment outcomes…”

– Added a section to the supplement titled “Appendix 3 – Antibiotic Resistance Analyses for Confirmed Failure and Relapse Patients” as well as new Appendix 3 – figure 1, Appendix 3—figure 2.

(3.2) The authors note a mutation frequency similar to the range derived in the absence of drug pressure (subsection “Characteristics of mutations in-host”) and refer in the Discussion section to inadequate therapy, which is hard to interpret with such heterogeneity. Due to the impact of baseline resistance (defined as allelic frequency >75%) and MDR disease on the development of new mutations during treatment, greater discussion of individual sample timing and duration/type of drug pressure would be helpful, as would be discussion of the allelic frequency threshold used to define prior resistance. Figure 6 confirms the significant impact of drug pressure as the primary driver of these mutations, rather than mutations in epitope encoding genes, so the extent to which mutations in epitope encoding genes are specifically varying in response to host activity (or not varying) is not clearly related to the host response from the data as presented.

We thank the reviewer for raising several points in this comment. First to clarify, baseline drug resistance was inferred from the whole genome sequencing data using a well validated set of 177 mutations at an allele frequency threshold (>40%). Selection of these mutations and validation of this allele frequency threshold was previously described (PMID 26910495 and cited 90 times to date). This study made use of 1,319 clinical Mtb isolates with known drug resistance phenotypes. The data were randomly split into training and validation sets containing 67% and 33% of the isolates (respectively). The diagnostic set of mutations was determined using random forest predictive modeling in which a weighted model was run with serially smaller subsets of mutations to identify a minimal set of mutations to predict resistance to first- and second-line TB drugs. The resulting set of mutations predicted INH resistance with a sensitivity of 94% and specificity of 94% on the validation isolate set and predicted RIF resistance with a sensitivity of 93% and specificity of 95% on the validation isolate set. In response to the reviewer comments we now provide further details in subsection (“Pre-existing Genotypic Resistance”, have updated Supplementary file 2 to include columns that indicated genotypic susceptibility or resistance to Isoniazid and Rifampicin in the first collected sample for all subjects) and have included Supplementary file 21 that contains the genotypic resistance predictions for 13 antibiotics for all 614 longitudinal isolates from the 307 patients in our study.

With regards to the point raised about the observed evolutionary mutation rate over time, we want to note that prior publications relied on serial samples from patients receiving antibiotic treatment and measured a similar rate (Walker, 2013 and personal communication with the author). The measured genome-wide mutation rate has been consistent between non-human primate Mtb evolutionary experiments (Ford et al., PMID: 21516081), Bayesian molecular clock estimation (Menardo, 2019,) and in-host pathogen evolution that largely come from patients receiving antibiotic treatment at least for some interval (Walker, PMID: 27701423). This is likely the case as drug exposure can result in selective pressure but in only a short section of the genome, and averaging mutation rates across all regions of the genome, as is done for mutation rate calculations, and the stochastic nature of mutation accumulation at this short time scale washes out this localized increase in diversity. We also refer the referee to Response 2.5 where we describe an analysis measuring the mutation rate in the subset of patients confirmed to have treatment failure.

With regards to the third point about lack of sample timing, we note that sample collection dates were available for all patients analysed 119 patients with confirmed failure and 195 total, with clonal infection. In response to the reviewer comments and previous comments from reviewer 1 and 2, we reran the assessment of new AR mutation development in the subset confirmed to be treatment failure and relapse, i.e. excluding cases of possible default for which treatment durations and regimens are not well documented. We measure a point estimate of AR of 9%, (95% *CI* [5.2%, 15.8%] *binconf* function in R) compared with the rate of 15% (95% *CI* [10.6%, 21.2%] *binconf* function in R) we observed across all persistent clonal infections We note the substantial overlap of the confidence intervals for these point estimates. The association between resistance acquisition and MDR also held among the 178/200 subset of with persistent or relapsed infection >2months (OR=3.85, *P* = 2.2x10-4 Fisher’s exact test) and among the 119/121 subjects with confirmed failure (OR=3.9, *P* = 4x10-2 Fisher’s exact test).

On the last point raised on in-host evolution of non-AR regions, we postulate that important drivers for evolution of other regions of the genome include host immunity and Mtb’s metabolic needs during chronic active infection. These selective forces may be independent of drug exposure and treatment regimen details as long as chronic infection, as evidenced by persistent bacterial growth from patient samples, is maintained. We specifically found evidence for selection on bacterial proteins involved with innate immune interactions and cobalamin biosynthesis proteins among other pathways described in the discussion. The study of these forces is very novel and has not been attempted previously at this scale, and we are the first to document evidence of non-AR based selection in host. We believe this is an important contribution to the literature on Mtb evolution and host-pathogen interactions more generally.

(3.3) Similarly, the authors note that mutations in drug resistance-associated loci is common and occurs across distinct Mtb lineages. While the finding of phylogenetic convergence is important, an alternative framing of the finding would be to consider these sites to vary in the presence of drug pressure independent of lineage.

We agree with the reviewer. Our test of phylogenetic convergence, the independent occurrence of certain mutations occurring in different genetic backgrounds, assesses whether a mutation arose multiple times independent of genetic background/lineage (annotated signature of positive selection). Our aim in this analysis was precisely the assessment of mutations and genes relevant to a phenotype independent of genetic background. This signal has previously been used to infer regions of positive selection in the context of antibiotic resistance (PMID: 23995135, cited 364 times). It may be possible that some background mutations or genes interact with newly acquired mutations to mediate the phenotype, but our approach is not designed to assess these situations.

(3.4) Of related concern, the authors defined study participants as having met criteria for failed therapy due to culture positivity after only 2 months of treatment. While patients receiving appropriate treatment ideally develop early culture conversion, many sources would require a longer time frame than 2 months to assign treatment failure, particularly if these participants were receiving experimental study regimens. Due to the heterogeneity introduced by the sample inclusion strategy, the conclusions of the manuscript with respect to negative treatment outcomes should be presented with greater weight placed on time between samples and time since starting therapy. Reframing the findings of the study as changes that occur during treatment, rather than changes occurring in the setting of failure, could also improve the support for the conclusions of the manuscript.

We thank the reviewer for raising this point. We refer the referee to Response 1.1 where we detail edits to the title and Responses 1.4 and 3.1 in which we detail how we generated a new and detailed patient treatment metadata Table (Supplementary file 2) that details treatment regimens received, and describes exactly when samples were collected relative to treatment initiation.

(3.5) For example, the authors identify a strong correlation between the SNP diversity of each participant's first sample and second sample and conclude that this demonstrates ineffective therapy (subsection “Determinants of antibiotic resistance acquisition and microbiological treatment failure”). How did this correlate among those participants who were not thought to have failed therapy? Did this vary with time to culture conversion among those converted later (that is, differentiating between those with "appropriate culture conversion", "delayed culture conversion," and true "failure"). A comparison between participants with eventual success and those who either never converted or who changed regimens due to emerging resistance would help differentiate this issue. Alternatively, a comparison between the SNP diversity among the 44 participants excluded due to different strains might better support the conclusion in the discussion that the sustained diversity identified is due to the absence of effective therapy.

We thank the reviewer for raising this point and we agree that these analyses would be insightful. Unfortunately, we are limited in the data available to us for the comparison groups. We only have 4 patients that are on effective therapy and the analysis of the SNP diversity between those patients’ first and second samples will be confounded by the much longer time duration of sampling for the failure cases. Furthermore, we don’t have data on time to culture conversion for our longitudinal samples, we only know that the majority were persistently culture positive at more than 2 months. A comparison between the SNP diversity between first and second samples among the 44 participants that have been classified as a reinfection (different strains) would be difficult to interpret as these populations are non-clonal and the diversity between them will be dominated by lineage or ancestral SNP differences.

(3.6) Finally, the Introduction is a bit confusing, combining discussions of the impact of host immunity, drug pressure, and microbiological and sequencing biases in selection of bacterial subpopulations. The result is that the reader is left confused about how to frame their interpretation of the study findings. This may be improved by a simpler introduction of the problem of unknown in-host variation over time and a summary of experiments that explain which bacterial factors should be considered at baseline to help predict future changes in Mtb isolates as well as the fixation of mutations in different genomic loci over time.

We thank the reviewer for their comment. We agree that the Introduction combined many discussions that interrupted the flow of the manuscript. To make the Introduction simpler and help with the flow, we have now revised it. The Introduction now flows from explaining the importance of studying the temporal dynamics of Mtb in-host, a short explanation of the importance in studying minor allele frequencies longitudinally for drug treatment, the barriers to studying the temporal dynamics of Mtb populations in-host using WGS, our snapshot of our sample and some of our main results.